# Activation of the CaMKII-Sarm1-ASK1-p38 MAP kinase pathway protects against axon degeneration caused by loss of mitochondria

Chen Ding[1], Youjun Wu[2], Hadas Dabas[2], Marc Hammarlund[1,2]*

[1]Department of Neuroscience, Yale University School of Medicine, New Haven, United States; [2]Department of Genetics, Yale University School of Medicine, New Haven, United States

**Abstract** Mitochondrial defects are tightly linked to axon degeneration, yet the underlying cellular mechanisms remain poorly understood. In *Caenorhabditis elegans,* PVQ axons that lack mitochondria degenerate spontaneously with age. Using an unbiased genetic screen, we found that cell-specific activation of CaMKII/UNC-43 suppresses axon degeneration due to loss of mitochondria. Unexpectedly, CaMKII/UNC-43 activates the conserved Sarm1/TIR-1-ASK1/NSY-1-p38 MAPK pathway and eventually the transcription factor CEBP-1 to protect against degeneration. In addition, we show that disrupting a trafficking complex composed of calsyntenin/CASY-1, Mint/LIN-10, and kinesin suppresses axon degeneration. Further analysis indicates that disruption of this trafficking complex activates the CaMKII-Sarm1-MAPK pathway through L-type voltage-gated calcium channels. Our findings identify CaMKII as a pivot point between mitochondrial defects and axon degeneration, describe how it is regulated, and uncover a surprising neuroprotective role for the Sarm1-p38 MAPK pathway in this context.

**\*For correspondence:**
marc.hammarlund@yale.edu

**Competing interest:** The authors declare that no competing interests exist.

## Editor's evaluation

This work reports exciting findings that show that activation of CaMKII downstream of the L-type Ca channel protects axons from RIC-7-induced axon degeneration. The authors also report the surprising result that the worm SARM, TIR-1 together with NSY-1 and SEK-1, mediates this protection effect.

## Introduction

Mitochondria are abundant in axons and at most synapses. The critical importance of these distal mitochondria is highlighted by the close relationship between defects in mitochondria localization or function and multiple neurodegenerative diseases. For some forms of hereditary neurodegeneration, such as hereditary spastic paraplegias, dominant optic atrophy, and Charcot–Marie–Tooth hereditary neuropathy type 2A (CMT2A), mutations in genes encoding distinct mitochondrial proteins are thought to be causative (*Alavi and Fuhrmann, 2013*; *Blackstone, 2018*; *Züchner et al., 2004*). Additionally, familial forms of Parkinson's disease (PD) arise from defective mitochondria quality control associated with mutations in the PINK1 and Parkin genes (*Pickrell and Youle, 2015*). Mitochondrial dysfunction has also been suggested to play roles in the more common, sporadic, forms of both PD and Alzheimer's disease (AD) (*Grünewald et al., 2019*; *Hauptmann et al., 2006*). However, the

**eLife digest** Within the cell are various compartments that carry out specific roles. This includes the mitochondria, which are responsible for generating the chemical energy that powers the cell. Some of the most power-hungry cells are nerve cells, which have long, slender projections called axons that relay signals from one part of the nervous system to another. If the mitochondria do not work properly, the axons break down which can lead to neurodegenerative diseases, such as Alzheimer's and Parkinson's disease.

Previous studies showed that defects in how mitochondria are transported cause axons in the roundworm *Caenorhabditis elegans* to spontaneously deteriorate with age. These transparent worms are often used to study biological questions related to the nervous system. Here, Ding et al. have used this model organism to identify a molecular pathway that stops axons from degenerating.

Random mutations were introduced into the genome of *C. elegans* that are unable to transport mitochondria in to their axons. Ding et al. then searched for mutant strains that still had intact axons despite this mitochondrial defect. This revealed that mutations that activate a protein called CaMKII stop axons from breaking down. Further experiments showed that CAMKII does this by switching on a series of signals, including the protein Sarm1, that eventually turn on another protein that suppresses degeneration.

The protective role of Sarm1 is surprising given that this protein has been shown to promote axon degeneration after injury in flies and mammals. This suggests that the gene for Sarm1 as well as others likely play different roles depending on the context in which they are activated.

These findings could help researchers identify new drug targets and strategies for treating neurodegenerative diseases caused by mitochondrial defects. However, further work is needed to see if this newly discovered pathway works the same way in other model systems, such as flies and mammals.

---

cellular mechanisms that determine the neurodegenerative response to mitochondria mislocalization or dysfunction are not well understood.

One well-established prodegenerative pathway in neurons involves dysregulated calcium homeostasis. For example, in neurodegenerative diseases including AD, PD, Huntington's disease (HD), and amyotrophic lateral sclerosis (ALS), elevated intracellular calcium levels have been observed. Proposed mechanisms resulting in elevated calcium include weakening of calcium buffering capacity, deregulation of calcium channel activities, and disruption of mitochondrial calcium homeostasis (*Bezprozvanny, 2009*; *Marambaud et al., 2009*; *Zündorf and Reiser, 2011*). Excess cytosolic calcium activates the calpain family of $Ca^{2+}$-dependent proteases, which cleave a variety of neuronal substrates and lead to degeneration (*Camins et al., 2006*). $Ca^{2+}$/calmodulin-dependent protein kinase II (CaMKII) has also been shown to mediate neurodegeneration in the context of elevated intracellular calcium levels (*Hou et al., 2009*; *Woolums et al., 2020*). In addition, elevated calcium causes mitochondrial malfunctions (*Bezprozvanny, 2009*). Mitochondria can buffer cytosolic calcium through the mitochondrial $Ca^{2+}$ uniporter (MCU) and can release their calcium content through the mitochondrial permeability transition pore (mPTP) (*Halestrap, 2009*; *Hoppe, 2010*). Mitochondria calcium overload, together with mitochondrial depolarization and oxidative stress, can trigger the opening of mPTP, which also leads to the release of apoptotic factors and triggers cell death (*Halestrap, 2009*). Thus, elevated calcium can promote neurodegeneration via various mechanisms.

More recently, a second major prodegenerative pathway in neurons was identified, centered on Sarm1. Sarm1 is essential for Wallerian degeneration in flies and mammals (*Gerdts et al., 2013*; *Osterloh et al., 2012*) and has a role in selected additional degeneration models such as traumatic brain injury (*Henninger et al., 2016*) and peripheral neuropathy (*Turkiew et al., 2017*). The Sarm1 TIR domains possess intrinsic $NAD^+$ hydrolase activity (*Essuman et al., 2017*), and activation of Sarm1 rapidly depletes $NAD^+$ in axons, leading to metabolic catastrophe and eventually axon degeneration (*Gerdts et al., 2015*). In addition to $NAD^+$ metabolism, Sarm1 has been shown to activate MAPK pathways (*Chuang and Bargmann, 2005*; *Hsu et al., 2021*; *Yang et al., 2015*). The role of MAPK signaling was examined in Wallerian degeneration, but mixed results were reported. Two studies showed that the c-Jun N-terminal kinase (JNK) MAPK pathway promotes axon degeneration, although its epistatic relationship with Sarm1 remains controversial (*Walker et al., 2017*; *Yang et al., 2015*). However, an

in vivo study in *Drosophila* argues against the requirement of JNK signaling in axon degeneration (*Neukomm et al., 2017*). Sarm1 also contains a mitochondrial targeting sequence (MTS) for association with the outer mitochondrial membrane (*Panneerselvam et al., 2012*). However, the importance of interactions between Sarm1 and mitochondria is still unclear as cytosolic Sarm1 protein lacking MTS is fully capable of activating degeneration and mitochondrial shape and motility is unaffected in Sarm1 KO neurons (*Gerdts et al., 2013*; *Summers et al., 2014*).

In this study, to identify cellular mechanisms that oppose or promote neurodegeneration in response to mitochondria dysfunction, we analyzed a *Caenorhabditis elegans* model of axon degeneration caused by complete absence of axonal mitochondria (*Rawson et al., 2014*). Using an unbiased approach, we discovered that $Ca^{2+}$/calmodulin-dependent protein kinase II (CaMKII/UNC-43) activity cell-autonomously protects against degeneration, even though mitochondria mislocalization persists. We found that CaMKII/UNC-43 abundance in axons is greatly reduced in the absence of mitochondria, suggesting that CaMKII/UNC-43 insufficiency is a key pivot point that links the loss of mitochondria to axon degeneration. Surprisingly, axon protection by CaMKII/UNC-43 requires Sarm1/TIR-1 and the downstream NSY-1-SEK-1-PMK-3-CEBP-1 mitogen-activated protein kinase (MAPK) pathway (*Chuang and Bargmann, 2005*; *Hayakawa et al., 2006*). Finally, we found that loss of the conserved axonal transport factors calsyntenin or Mint suppresses axon degeneration, and that this suppression is dependent on L-type voltage-gated calcium channel (VGCC) and CaMKII/UNC-43. Together, our data provide insight into the cellular mechanisms that control the neurodegenerative response to mitochondria mislocalization and reveal a novel anti-degenerative function for Sarm1, calcium, and CaMKII.

## Results

### An unbiased screen identifies suppressors of axon degeneration caused by lack of mitochondria

In *C. elegans*, anterograde traffic of mitochondria from neuronal cell bodies to axons is known to depend on a small number of factors: *unc-116*, which encodes the kinesin heavy chain (KHC) of the kinesin-1 motor complex (*Schwarz, 2013*); the novel factor *ric-7*, which is specifically required for mitochondria localization in axons (*Rawson et al., 2014*); and the combined activity of Miro, Milton, and metaxin, which couple mitochondria to kinesin-1-mediated traffic (*Glater et al., 2006*; *Guo et al., 2005*; *Stowers et al., 2002*; *Zhao et al., 2021*). The two PVQ neurons are a pair of unipolar neurons in the posterior of the animal that each extends a single long neurite to the nerve ring in the head (*Figure 1A*). PVQ was previously shown to degenerate as a result of lack of axonal mitochondria in *ric-7* mutants (*Rawson et al., 2014*). We labeled both PVQ neurons with the *oyIs14[sra-6p::GFP]* transgene (which additionally marks the ASI and ASH neurons in the head) (*Nolan et al., 2002*; *Figure 1C*). We also labeled mitochondria in PVQ using a TagRFP marker targeted to the mitochondria matrix (mito::TagRFP) with the mitochondrial import signal from chicken aspartate aminotransferase (*Figure 1E*; *Ghose et al., 2013*; *Jaussi et al., 1985*). We quantified degeneration by determining the fraction of animals with no PVQ degeneration (two intact nontruncated PVQ axons) (*Figure 1B*). Consistent with previous findings (*Rawson et al., 2014*), loss of *ric-7* results in a complete absence of mitochondria in PVQ axons as early as the L1 stage, which continues into and past the L4 stage (*Figure 1E–G*). The loss of axonal mitochondria is accompanied by completely penetrant degeneration of PVQ axons in adults (*Figure 1C and D*). Most axons degenerate during the L4 to 1-day-old adult (1doa) transition (*Figure 1D*). Similarly, PVQ axons also degenerate in KHC/*unc-116* mutants and Miro-metaxin/*miro-1; mtx-2* double mutants (*Figure 1—figure supplement 1A*), in which mitochondria are completely absent in axons (*Figure 1—figure supplement 1B*; *Rawson et al., 2014*; *Zhao et al., 2021*). By contrast, PVQ axons do not degenerate in *miro-1; miro-2* double mutants or in Milton/*trak-1* single mutants (*Figure 1—figure supplement 1A*), likely because mitochondria are still present in axons in these mutants (*Figure 1—figure supplement 1B*; *Sure et al., 2018*; *Zhao et al., 2021*). Therefore, spontaneous PVQ axon degeneration is induced by the complete loss of axonal mitochondria. Because kinesin-1 is involved in the transport of multiple axonal cargoes in addition to mitochondria (*Guedes-Dias and Holzbaur, 2019*), we used *ric-7* mutants for further analysis of the relationship between axonal mitochondria and degeneration.

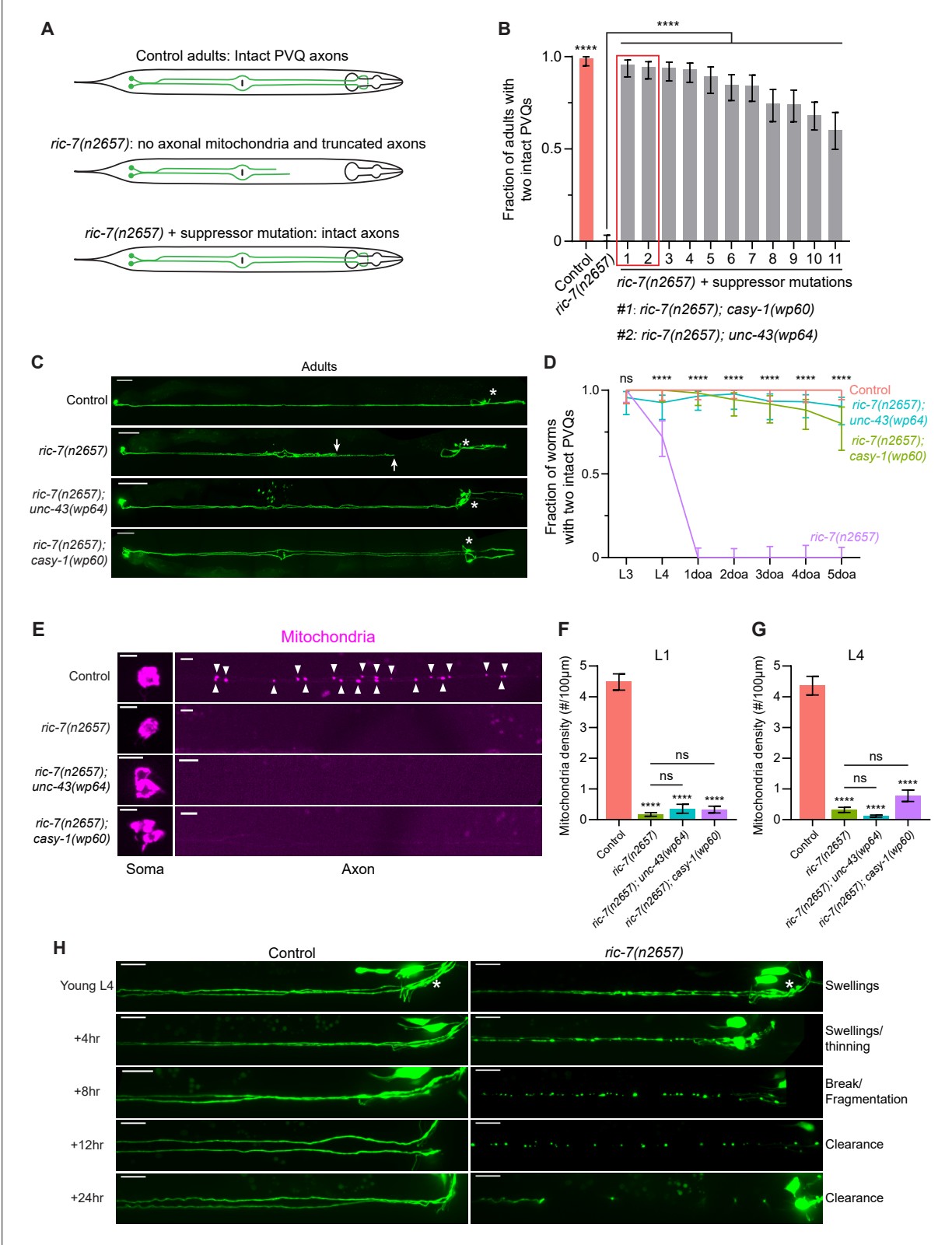

**Figure 1.** An unbiased screen identifies suppressors of spontaneous axon degeneration caused by loss of mitochondria. (**A**) Diagrams of PVQ neurons in control, *ric-7(n2657)*, and *ric-7(n2657)* + suppressor animals. Throughout this study, PVQ neurons are visualized with *oyIs14[sra-6p::GFP]*, and mitochondria in PVQ are visualized with *sra-6p::mito::TagRFP*. (**B**) Axon degeneration modifiers identified in unbiased forward genetic screen. Graph shows proportion of 3-day-old adult (3doa) animals without degeneration in either PVQ neuron. Leftmost bars show control (99% animals without

*Figure 1 continued on next page*

*Figure 1 continued*

degeneration) and *ric-7(n2657)* (0% animals without degeneration). Remaining bars show strains that are all *ric-7(n2657)* mutants and that also carry an independent mutation that suppresses degeneration. The top two suppressor mutants, *ric-7(n2657); casy-1(wp60)* and *ric-7(n2657); unc-43(wp64)*, are highlighted. Bars show proportion and 95% confidence intervals (CIs), N > 74 for all strains. ****p<0.0001, compared to *ric-7(n2657)*, Fisher's exact test. (**C**) Suppression of axon degeneration by mutations in CaMKII/*unc-43* and calsyntenin/*casy-1*. Representative images of control, *ric-7(n2657)*, *ric-7(n2657); unc-43(wp64)*, and *ric-7(n2657); casy-1(wp60)* 2–3doa. Arrows indicate the tips of degenerated axons. Asterisks indicate the head neurons— these are co-labeled by the GFP reporter but are not the subject of this study. Scale bar, 50 µm. (**D**) Quantification of suppression of axon degeneration. PVQ degeneration is analyzed from the L3 stage to 5-day-old adult (5doa) in control and suppressor mutants. Graph shows proportion and 95% CI. N = 44–74 for each timepoint. ****p<0.0001, ns, not significant, Chi-square test. (**E**) Suppressors of degeneration do not restore axonal mitochondria. Images of mitochondria in PVQ neurons in control, *ric-7(n2657)*, and suppressor mutants at L4. Arrowheads indicate mitochondria in axons. Scale bar, 5 µm. (**F, G**) Quantification of mitochondria density in control, *ric-7(n2657)*, and suppressor mutants at L1 and L4 stages. Bars show mean and SEM. N = 16, 26, 63, 53 for L1s and N = 54, 52, 50, 70 for L4s. ****p<0.0001, ns, not significant, compared to control except where indicated, one-way ANOVA, Kruskal–Wallis test, followed by Dunn's multiple comparisons. (**H**) Axon degeneration in the absence of mitochondria is progressive. Images of axon morphology in a single control and a single *ric-7(n2657)* animal at timepoints during the L4-1doa transition. Asterisk indicates the nerve ring. Scale bar, 10 µm.

The online version of this article includes the following source data and figure supplement(s) for figure 1:

**Source data 1.** Axon degeneration and mitochondria density in control and suppressor mutants.

**Figure supplement 1.** Complete loss of axonal mitochondria leads to degeneration and key allele information.

**Figure supplement 1—source data 1.** Axon degeneration and mitochondria density in mutants that affect mitochondria transport.

**Figure supplement 2.** *casy-1(wp60)* and *unc-43(wp64)* do not suppress enhanced injury-induced axon degeneration (after laser axotomy) in *ric-7(n2657)*.

**Figure supplement 2—source data 1.** Axotomy-induced axon degeneration in *ric-7*, *ric-7; casy-1* and *ric-7; unc-43 gof* animals.

**Figure supplement 3.** *casy-1(wp78)* and *unc-43(n498sd)* suppress axon degeneration in *miro-1(wy50180); mtx-2(wy50266)* mutants.

**Figure supplement 3—source data 1.** Axon degeneration in *miro-1; mtx-2*, *miro-1; mtx-2; casy-1* and *miro-1; mtx-2; unc-43 gof* animals.

End-stage PVQ degeneration is visible in *ric-7* mutants as truncated axons that are disconnected from the nerve ring (**Figure 1A–C**). To determine the sequence of events that leads to degeneration, we performed longitudinal imaging of individual axons. We found that degeneration is a sequential process beginning with axon swelling, progressing through thinning and fragmentation, and ending with debris clearance (**Figure 1H**). To further confirm that PVQ axon degeneration in *ric-7* mutants is caused by the absence of mitochondria, we expressed a chimeric construct consisting of the UNC-116/kinesin-1 motor and the mitochondrial outer membrane protein TOMM-7, which restores axonal mitochondria localization, and we found that degeneration could be completely suppressed (**Figure 1—figure supplement 1C and D**; *Nichols et al., 2016*; *Rawson et al., 2014*). Thus, the PVQ axon in *ric-7* mutants is a completely penetrant system for analyzing axon degeneration caused by loss of axonal mitochondria.

To identify molecular mechanisms required for axon degeneration in the absence of mitochondria, we performed an unbiased suppressor screen in the *ric-7(n2657)* background (**Figure 1A**). After standard EMS mutagenesis and screening of semi-clonal F2s (see Materials and methods), we recovered 11 strains with >50% suppression of degeneration (**Figure 1B**). We cloned the two strongest suppressors, *wp64* and *wp60*, using whole-genome sequencing (see Materials and methods). *wp64* is an allele of *unc-43*, the *C. elegans* homolog of CaMKII, and *wp60* is an allele of *casy-1*, the *C. elegans* homolog of calsyntenin (**Figure 1B**, **Figure 1—figure supplement 1E**). In *ric-7(n2657); unc-43(wp64)* and *ric-7(n2657); casy-1(wp60)* animals, over 80% of 5-day-old adult (5doa) have no degeneration—both PVQ axons are intact (**Figure 1C and D**). However, despite the strong suppression of degeneration, mitochondria are still absent from axons in both suppressed strains at both the L1 and L4 stages (**Figure 1E–G**). Therefore, instead of restoring mitochondria localization in axons, the CaMKII/*unc-43(wp64)* and calsyntenin/*casy-1(wp60)* mutations maintain axon integrity in the absence of mitochondria.

## Constitutively active CaMKII/UNC-43 suppresses spontaneous axon degeneration cell-autonomously

The CaMKII/*unc-43(wp64)* allele is a G-to-A point mutation, which leads to a glycine to arginine substitution (G207R) in one of the two hydrophobic pockets of the catalytic domain (**Figure 1—figure supplement 1E**; *Umemura et al., 2005*). This hydrophobic pocket interacts with T284 on the autoinhibitory domain and is critical for the autoinhibition of CaMKII in low calcium (*Yang and*

*Schulman, 1999*). Abolishing this interaction with a T284D mutation in the autoinhibitory domain, or with a V206E mutation in the hydrophobic pocket, causes CaMKII/UNC-43 to be constitutively active (CA) even in the absence of calcium (*Umemura et al., 2005*; *Yang and Schulman, 1999*). Since the G207R alteration in our *wp64* allele is next to the V206E change that results in constitutive activity, we hypothesized that the *wp64* mutation also makes CaMKII/UNC-43 CA. Indeed, the *unc-43(wp64)* worms display a dominant lethargic phenotype, similar to the canonical CA *unc-43(n498sd)* mutants (*Reiner et al., 1999*). Further, we found that the *unc-43(n498sd)* mutation phenocopies *unc-43(wp64)* in terms of the anti-degeneration effect in *ric-7(n2657)* animals (*Figure 2A*). By contrast, the loss-of-function (lof) allele *unc-43(e408)* does not suppress degeneration in *ric-7(n2657)* animals (*Figure 2A*). Interestingly, although PVQ axons rarely degenerate when CaMKII/UNC-43 is activated, aberrant axonal swellings are still observed (*Figure 2B*), suggesting that degeneration is blocked at a step between initiation and severing. Mitochondria anterograde transport also requires kinesin-1/UNC-116 (*Hollenbeck and Saxton, 2005*), and adaptor proteins including MIRO-1 and MTX-2 (*Guo et al., 2005*; *Zhao et al., 2021*). PVQ axons spontaneously degenerate in *unc-116(rh24sb79)* animals between L3 and L4 (*Figure 2C*). We found that CA *unc-43(n498sd)* suppresses degeneration in *unc-116(rh24sb79)* (*Figure 2C*). Similarly, *unc-43(n498sd)* also suppresses degeneration in *miro-1(wy50180); mtx-2(wy50266)* double mutants (*Figure 1—figure supplement 3*). In conclusion, activated CaMKII/UNC-43 suppresses axon degeneration caused by the absence of mitochondria.

To examine if CaMKII/UNC-43 functions in PVQ cell-autonomously, we overexpressed CaMKII/UNC-43 in PVQ in *ric-7(n2657)* animals. Expressing wildtype CaMKII/UNC-43 leads to mild but distinct protection against degeneration (20% of 3doa have no degeneration), while expressing a CA form of CaMKII/UNC-43 (T284D) results in much stronger protection (70% of 3doa have no degeneration) (*Figure 2A*). These data indicate that active CaMKII/UNC-43 functions cell-autonomously to suppress axon degeneration. Further, since overexpression of wildtype CaMKII/UNC-43 has less effect on degeneration than expression of the CA protein, our results suggest that WT UNC-43 cannot be adequately activated even when overexpressed in PVQ axons without mitochondria.

To explore potential mechanisms that might affect CaMKII/UNC-43 function in animals lacking axonal mitochondria, we examined the abundance and distribution of CaMKII/UNC-43 protein using the native and tissue-specific fluorescence (NATF) approach (*He et al., 2019*). Briefly, seven copies of the GFP11 β-strand were knocked into the C terminus of the endogenous CaMKII/*unc-43* locus by CRISPR/Cas9. The complementary split-GFP fragment GFP1-10 was then expressed under the PVQ promoter, resulting in PVQ-specific labeling of CaMKII/UNC-43 at endogenous expression levels (*Figure 2D*). The *unc-43::gfp11x7* knock-in (KI) animals displayed no apparent uncoordinated or lethargic phenotype related to *unc-43* mutants (data not shown), indicating that CaMKII/UNC-43 function is not impaired by the KI. As a negative control, expressing GFP1-10 alone in PVQ did not produce visible GFP fluorescence (*Figure 2—figure supplement 1A*). We examined the distribution of labeled CaMKII/UNC-43 in control and *ric-7* animals at the L3 stage, which precedes spontaneous degeneration (*Figure 1D*; *Rawson et al., 2014*). We found that in *ric-7(n2657)* animals CaMKII/UNC-43 abundance in distal axons (posterior to the nerve ring) is dramatically reduced (37% of the control levels), while CaMKII/UNC-43 abundance in cell bodies is not affected (*Figure 2D and E*). Further, axonal CaMKII/UNC-43 abundance in *ric-7(n2657)* animals was partially restored (to 69% of control levels) by expression of the UNC-116::TOMM-7 chimeric protein, a synthetic method of forcing mitochondria into the axon in the absence of *ric-7* (*Rawson et al., 2014*; *Figure 2D and E*, *Figure 1—figure supplement 1C*). Therefore, our data indicate that axonal mitochondria promote CaMKII/UNC-43 localization in axons.

## CaMKII/UNC-43 suppresses axon degeneration through the conserved Sarm1/TIR-1-ASK1/NSY-1-PMK-3-CEBP-1 MAPK pathway

We next sought to identify the downstream targets of CaMKII/UNC-43. In *C. elegans* nervous system, CaMKII/UNC-43 has been shown to regulate asymmetric neural development through the conserved Sarm1/TIR-1-ASK1/NSY-1-p38 MAPK pathway (*Chuang and Bargmann, 2005*; *Hayakawa et al., 2006*; *Figure 2F*). Specifically, in one of the two AWC sensory neurons, high calcium activates UNC-43, which then directly interacts with TIR-1 and releases the autoinhibition on the TIR domains. Active TIR-1 then recruits and activates the NSY-1 MAPKKK to inhibit the expression of specific odorant receptors. In addition, UNC-43, TIR-1, and NSY-1 colocalize at synaptic regions of AWC and physically

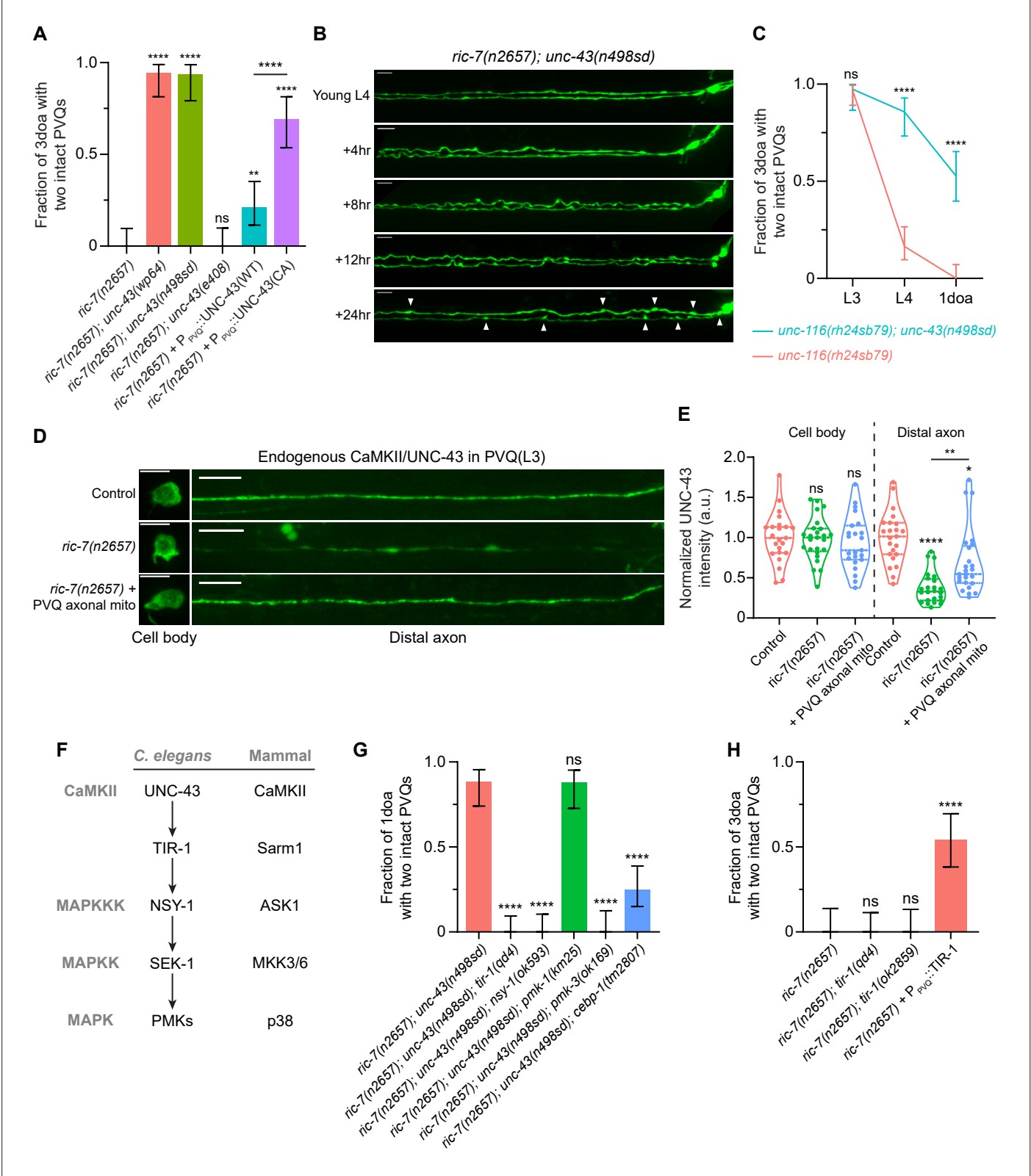

**Figure 2.** Active CaMKII/UNC-43 suppresses axon degeneration cell-autonomously through the conserved Sarm1/TIR-1-ASK1/NSY-1 MAPK pathway. (**A**) Active CaMKII/UNC-43 functions cell-autonomously to suppress degeneration. Quantification of axon degeneration in 3-day-old adult (3doa) animals. Genotypes and number of animals: *ric-7(n2657)* (N = 36), *ric-7(n2657); unc-43(wp64)* (N = 35), *ric-7(n2657); unc-43(n498sd)* (N = 31), *ric-7(n2657); unc-43(e408)* (N = 35), *ric-7(n2657)* + P$_{PVQ}$::WT UNC-43 (N = 43) and *ric-7(n2657)* + P$_{PVQ}$::constitutively active (CA) UNC-43 (N = 39). Bars show proportion

*Figure 2 continued on next page*

*Figure 2 continued*

and 95% CI. ****p<0.0001, **p<0.01, ns, not significant, compared to ric-7(n2657) except where indicated, Fisher's exact test. (**B**) Images of axon morphology in a single *ric-7(n2657); unc-43(n498sd)* animal at timepoints during the L4-1doa transition. Arrowheads indicate axon swellings. Scale bar, 10 µm. (**C**) Loss of kinesin-1/*unc-116* results in axon degeneration that is suppressed by active CaMKII/*unc-43*. Quantification of axon degeneration from the L3 stage to 1-day-old adults (1doa). Genotypes and number of animals: *unc-116(rh24sb79)* (N = 50–73), and *unc-116(rh24sb79); unc-43(n498sd)* (N = 38–55). Graph shows proportion and 95% CI. ****p<0.0001, ns, not significant, compared to *unc-116(rh24sb79)*, Fisher's exact test. (**D**) Endogenous CaMKII/UNC-43 localization. Images of native and tissue-specific fluorescence (NATF)-tagged CaMKII/UNC-43 in PVQ cell bodies and distal axons adjacent to the nerve ring in control, *ric-7(n2657)*, and *ric-7(n2657)* + P~PVQ~::UNC-116::TOMM-7 in L3 stage animals. Scale bar, 5 µm. (**E**) Normalized NATF-tagged CaMKII/UNC-43 intensities (arbitrary units) in PVQ cell bodies and distal axons in control, *ric-7(n2657)* and *ric-7(n2657)* + P~PVQ~::UNC-116::TOMM-7 in L3 stage animals. Violin plots with median and quantiles are shown. ****p<0.0001, **p<0.01, *p<0.05, ns, not significant, compared to control except where indicated, one-way ANOVA, Kruskal–Wallis test, followed by Dunn's multiple comparisons. (**F**) The UNC-43-TIR-1-MAPK pathway and conservation in mammals. (**G**) The TIR-1-NSY-1-SEK-1-PMK-3-CEBP-1 pathway is required to suppress axon degeneration in activated CaMKII/*unc-43* mutants. Quantification of axon degeneration in 1doa. Genotypes and number of animals: *ric-7(n2657); unc-43(n498sd)* (N = 35), *ric-7(n2657); unc-43(n498sd); tir-1(qd4)* (N = 37), *ric-7(n2657); unc-43(n498sd); nsy-1(ok593)* (N = 33), *ric-7(n2657); unc-43(n498sd), pmk-1(km25)* (N = 33), *ric-7(n2657); unc-43(n498sd), pmk-3(ok169)* (N = 27), and *ric-7(n2657); unc-43(n498sd), cebp-1(tm2807)* (N = 48). Bars show proportion and 95% CI. ****p<0.0001, ns, not significant, compared to *ric-7(n2657); unc-43(n498sd)*, Fisher's exact test. (**H**) Sarm1/TIR-1 protects against degeneration in the context of lack of axonal mitochondria. Quantification of axon degeneration in 3doa. Genotypes and number of animals: *ric-7(n2657)* (N = 24), *ric-7(n2657); tir-1(qd4)* (N = 30), *ric-7(n2657); tir-1(ok2859)* (N = 25) and *ric-7(n2657)* + P~PVQ~::WT TIR-1 (N = 35). Bars show proportion and 95% CI. ****p<0.0001, ns, not significant, compared to *ric-7(n2657)*, Fisher's exact test.

The online version of this article includes the following source data and figure supplement(s) for figure 2:

**Source data 1.** Axon degeneration in mutants in the CaMKII-Sarm1-MAPK pathway and endogenous CaMKII abundance.

**Figure supplement 1.** The GFP1-10 control and the *casy-1::gfp11x7* control for the native and tissue-specific fluorescence (NATF) approach.

**Figure supplement 1—source data 1.** The *gfp11x7* insertion at the *casy-1* locus does not affect axon degeneration.

**Figure supplement 2.** Overexpressing worm NMNATs does not suppress degeneration in *ric-7(n2657)*.

**Figure supplement 2—source data 1.** Axon degeneration in *ric-7* animals that overexpress NMATs in PVQ.

interact with each other to form a signaling complex (*Chuang and Bargmann, 2005*; *Sagasti et al., 2001*). To test if this signaling complex functions in PVQ degeneration, we examined whether the individual components were required for the protection against degeneration conferred by *unc-43(n498sd)* in animals lacking axonal mitochondria. We found that lof mutations in *tir-1(qd4)* and *nsy-1(ok593)* completely abolish the protection against degeneration conferred by the CA *unc-43(n498sd)* allele (*Figure 2G*). Thus, the UNC-43/TIR-1/NSY-1 complex functions to suppress axon degeneration in the absence of mitochondria. During development, this complex regulates asymmetric neuronal development via the MAPKK SEK-1 and two redundant p38 MAPKs, PMK-1 and PMK-2 (*Alqadah et al., 2016b*; *Pagano et al., 2015*). We could not assess SEK-1 function in axon degeneration, as we found that *ric-7(n2657); sek-1(km4); unc-43(n498sd)* animals are sick and arrest before adulthood, precluding analysis of degeneration. However, we found that loss of PMK-1 alone does not prevent activated CaMKII/UNC-43 from suppressing degeneration (*Figure 2G*). By contrast, loss of another p38 MAPK, PMK-3, or loss of its downstream C/EBP bZIP transcription factor CEBP-1 (*Yan et al., 2009*) largely blocks the protective effects of activated CaMKII (*Figure 2G*). Therefore, active CaMKII protects against axon degeneration through CEBP-1-mediated transcription regulation.

Our data suggest that Sarm1/TIR-1 has an axon protective role: it is required for constitutively activated CaMKII to suppress degeneration due to mitochondria mislocalization (*Figure 2I*). By contrast, Sarm1/TIR-1 in other contexts—particularly Wallerian degeneration—promotes axon degeneration by depleting axonal NAD⁺ (*Gerdts et al., 2013*; *Henninger et al., 2016*; *Osterloh et al., 2012*; *Turkiew et al., 2017*). We examined whether loss of Sarm1/TIR-1 might suppress degeneration in axons that lack mitochondria. We found that neither of the two lof mutations of *tir-1*, *qd4* and *ok2859*, suppresses PVQ degeneration (*Figure 2J*). Similarly, overexpressing the *C. elegans* NAD⁺ synthase *nmat-1* or *nmat-2* does not suppress degeneration in *ric-7(n2657)* (*Figure 2—figure supplement 2*). By contrast, overexpression of WT TIR-1 in PVQ strongly suppresses degeneration (*Figure 2J*). Overall, our results indicate a surprising neuroprotective role for Sarm1/TIR-1 as part of the UNC-43/TIR-1/NSY-1 signaling complex.

## Loss of calsyntenin/CASY-1 suppresses axon degeneration

Our unbiased screen identified a mutation in the *casy-1* gene, *wp60*, that suppresses PVQ degeneration to a similar extent as activated CaMKII/UNC-43 (*Figure 3A and B*). CASY-1 is the *C. elegans*

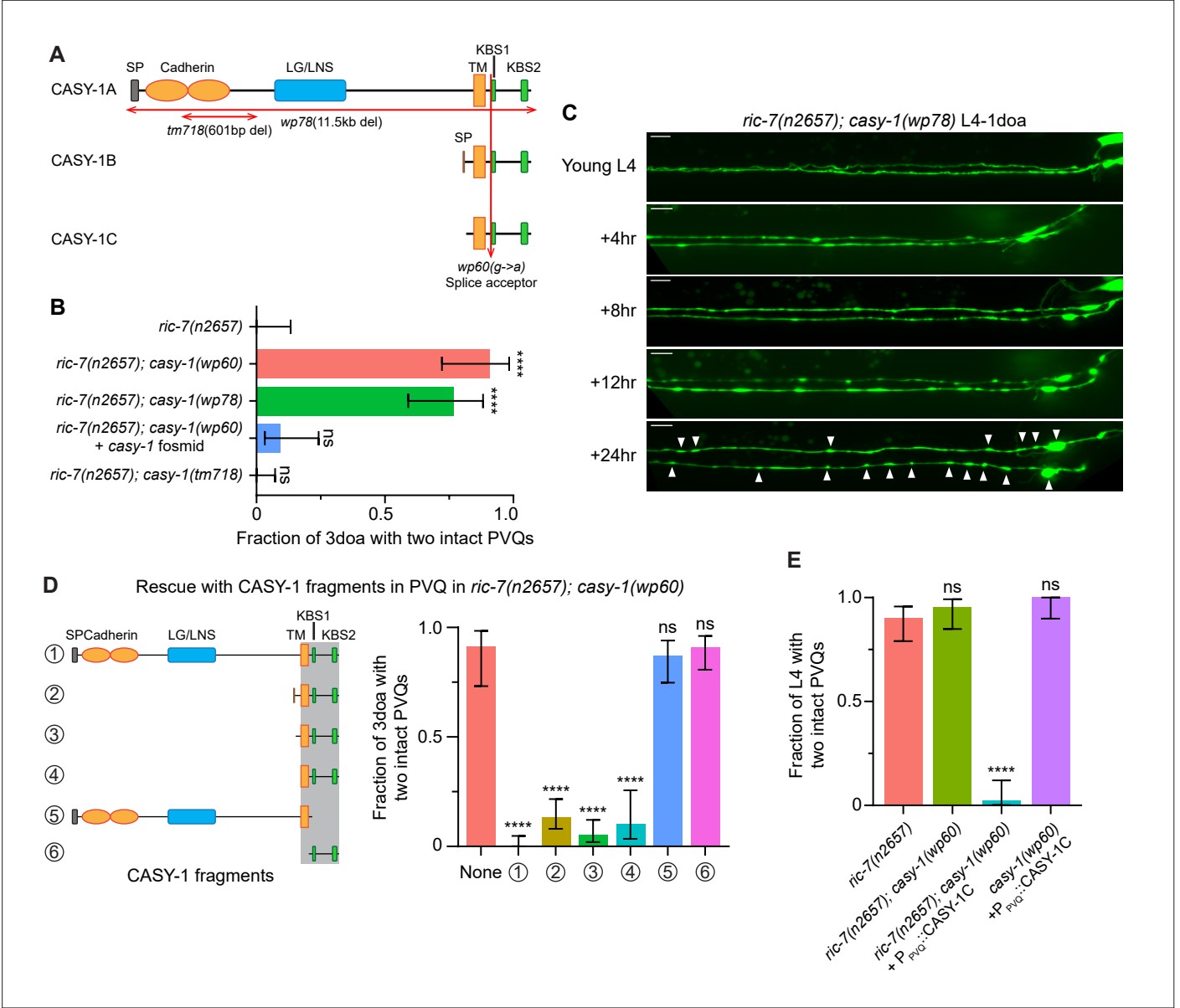

**Figure 3.** Calsyntenin/CASY-1 promotes axon degeneration cell-autonomously. (**A**) The three *C. elegans* CASY-1 isoforms. SP, signal peptide; LG/LNS, laminin G-like; TM, transmembrane domain; KBS, kinesin-binding site. (**B**) Loss of calsyntenin/*casy-1* suppresses degeneration due to loss of axonal mitochondria. Quantification of axon degeneration in 3-day-old adult (3doa). Genotypes and number of animals: *ric-7(n2657)* (N = 25), *ric-7(n2657); casy-1(wp60)* (N = 22), *ric-7(n2657); casy-1(wp78)* (N = 30), *ric-7(n2657); casy-1(wp60)* + the *casy-1* fosmid WRM0622dH03 (N = 32), and *ric-7(n2657); casy-1(tm718)* (N = 49). Bars show proportion and 95% CI. ****p<0.0001, ns, =not significant, compared to *ric-7(n2657)*, Fisher's exact test. (**C**) Loss of calsyntenin/*casy-1* arrests degeneration before axon severing. Images of axon morphology in a *ric-7(n2657); casy-1(wp78)* animal during the L4-1doa transition. Arrowheads indicate axon swellings. Scale bar, 10 μm. (**D**) Structure-function analysis of CASY-1. Left, six CASY-1 fragments expressed in PVQ in *ric-7(n2657); casy-1(wp60)* animals. Right, quantification of axon degeneration in 3doa. Number of animals in experiments, respectively: 23, 76, 97, 80, 30, 47, and 56. Bars show proportion and 95% CI. ****p<0.0001, ns, not significant, compared to *ric-7(n2657); casy-1(wp60)* with no transgene, Fisher's exact test. Transgenic fragments #1–4 restore degeneration to *ric-7(n2657); casy-1(wp60)* mutants. (**E**) Overexpression of calsyntenin/*casy-1* enhances axon degeneration due to loss of axonal mitochondria. Quantification of axon degeneration in L4 stage animals. Genotypes and number of animals: *ric-7(n2657)* (N = 51), *ric-7(n2657); casy-1(wp60)* (N = 44), *ric-7(n2657); casy-1(wp60)* + P_PVQ::CASY-1C (N = 43), and *casy-1(wp60)* + P_PVQ::CASY-1C (N = 34). Bars show proportion and 95% CI. ****p<0.0001, ns, not significant, compared to *ric-7(n2657)*, Fisher's exact test.

The online version of this article includes the following source data for figure 3:

**Source data 1.** Axon degeneration in *casy-1* loss-of-function and overexpression animals.

homologue of calsyntenin, and comparison of CASY-1 to its *Drosophila* and mammalian homologues shows that these proteins share significant sequence similarity along their entire lengths (*Hill et al., 2001*). Calsyntenins are type I transmembrane (TM) proteins and are members of the cadherin superfamily (*Hill et al., 2001*). Their extracellular region contains two cadherin domains and an LG/LNS (laminin G-like) domain and has been shown to regulate synapse development through interactions with neurexin (*Kim et al., 2020a*; *Kim et al., 2020b*; *Pettem et al., 2013*). Their intracellular region is characterized by two kinesin-binding sites (KBS), which are important for its anterograde trafficking (*Araki et al., 2003*; *Konecna et al., 2006*).

The *C. elegans casy-1* locus encodes three isoforms (*Figure 3A*). The longest isoform, *casy-1a*, encodes a protein with all domains. The two shorter isoforms, *casy-1b and casy-1c*, share the TM domain and the intracellular region, but do not contain the extracellular cell adhesion domains. The *wp60* allele we identified in the screen is a G-to-A point mutation at the splice acceptor site in the intron before the first KBS, and it affects all three isoforms (*Figure 3A*, *Figure 1—figure supplement 1E*). We used CRISPR/Cas9 to delete the entire coding region of *casy-1* (*Figure 3A*, *Figure 1—figure supplement 1E*). This 11.5 kb deletion allele, *wp78*, phenocopies the *wp60* mutation from the screen and strongly suppresses PVQ degeneration in both *ric-7(n2657)* and *miro-1(wy50180); mtx-2(wy50266)* mutants (*Figure 3B*, *Figure 1—figure supplement 3*). In the *ric-7(n2657); casy-1(wp78)* animals, PVQ axons rarely degenerate and only develop a few swellings during the L4-1doa transition (*Figure 3C*). The phenotype is very similar to the protection conferred by activated CaMKII/UNC-43 (*Figure 2B*). Further, protection of *casy-1(wp60)* can be abolished by expressing a 34.6 kb fosmid that contains the endogenous *casy-1* locus (*Figure 3C*). Together, these data indicate that loss of calsyntenin/*casy-1* prevents degeneration of axons caused by lack of mitochondria.

## Calsyntenin/CASY-1 TM domain and ICD function together to promote axon degeneration cell-autonomously

To determine if CASY-1 acts cell-autonomously to promote axon degeneration, we performed PVQ-specific rescue experiments in *ric-7(n2657); casy-1(wp60)* animals, in which degeneration is suppressed due to loss of calsyntenin. Overexpression of full-length calsyntenin completely restored degeneration to *ric-7(n2657); casy-1(wp60)* animals, indicating that calsyntenin is required cell-autonomously for PVQ axon degeneration (*Figure 3D*). Next, we used this approach to define a minimal fragment of calsyntenin capable of supporting its pro-degeneration function. We examined six CASY-1 fragments (*Figure 3D*) for their abilities to promote axon degeneration in this context, including the three endogenous isoforms (#1–3, see also *Figure 3A*), a fragment containing the TM domain+ ICD (#4), CASY-1a without the C terminus starting from the first KBS (#5), and the ICD alone (#6). Expression of CASY-1 fragments #1–4, but not #5 or #6, caused strong PVQ degeneration in *ric-7(n2657); casy-1(wp60)* animals (*Figure 3D*). This structure-function analysis indicated that both calsyntenin's TM domain and the ICD are required for degeneration, while its extracellular region is dispensable. Consistent with these results, we did not observe suppression of degeneration in an existing *casy-1* allele, *tm718,* which is a deletion within the extracellular domain (*Figure 3A and B*).

We then examined if overexpressing *casy-1* promotes degeneration. Overexpressing CASY-1C in *ric-7(n2657); casy-1(wp60)* leads to near complete degeneration at L4, a timepoint at which only a few *ric-7(n2657)* control animals show degeneration (*Figure 3E*). However, overexpressing CASY-1C in *casy-1(wp60)* single mutants does not induce degeneration (*Figure 3E*). Therefore, overexpressing CASY-1 alone is not sufficient to cause degeneration, but is able to enhance degeneration in the absence of mitochondria.

## Anterograde traffic of calsyntenin/CASY-1 promotes axon degeneration

The calsyntenin ICD contains two conserved KBS, and in *C. elegans*, the CASY-1 intracellular region binds to kinesin light chain 2 (KLC-2) of the kinesin complex (*Araki et al., 2007*; *Ohno et al., 2014*). To test the importance of the KBS for degeneration, we mutated the conserved tryptophan residues in each KBS to alanine (*Figure 4—figure supplement 1A*; *Konecna et al., 2006*). We found that this change completely eliminates the pro-degeneration function of calsyntenin (*Figure 4A*). These data suggest that calsyntenin functions together with kinesin to mediate degeneration. Next, we tested whether loss of KLC-2 would suppress degeneration, similar to loss of calsyntenin. Indeed, the *klc-2(km11)* lof mutation strongly suppresses degeneration in axons that lack mitochondria (*Figure 4B*).

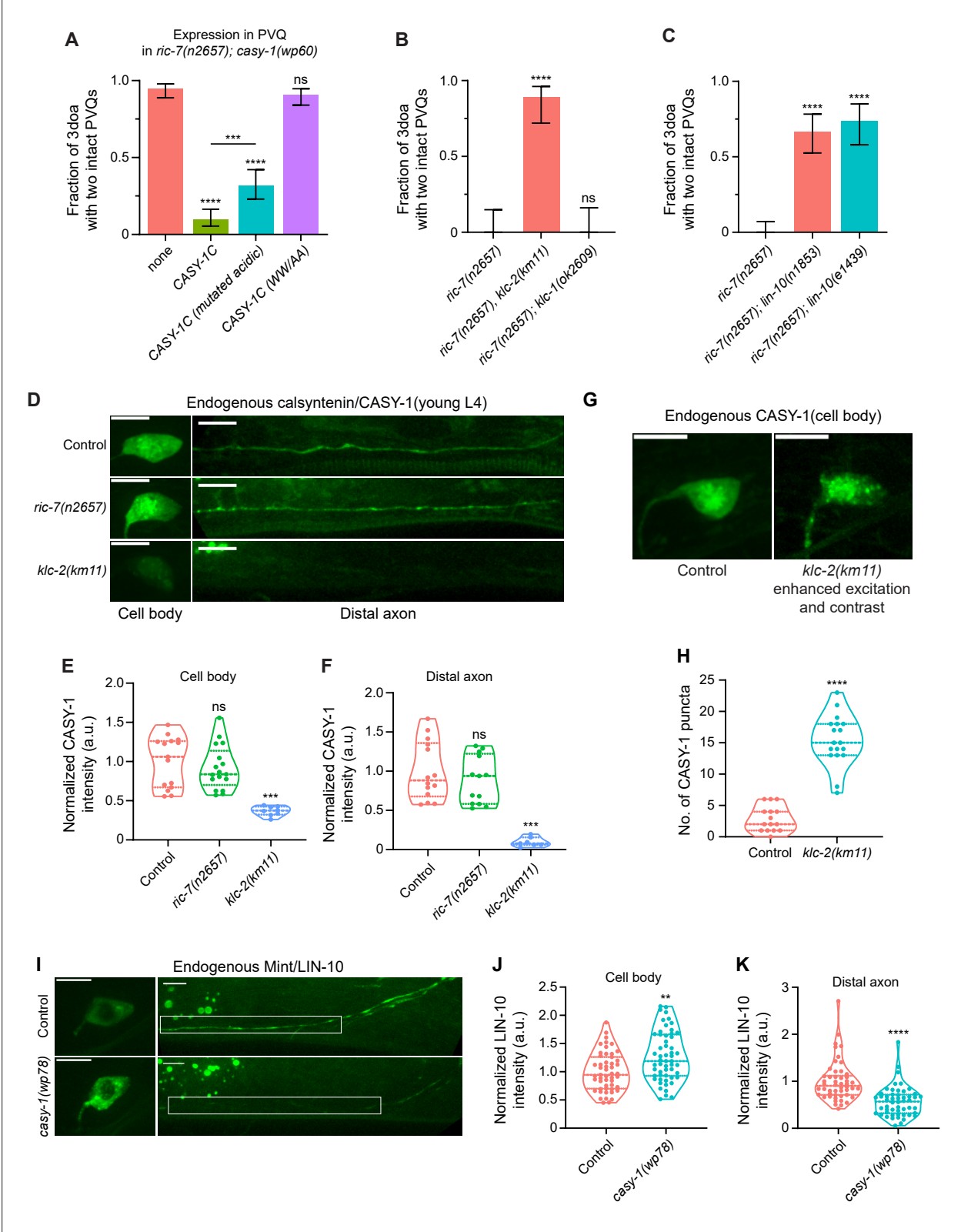

**Figure 4.** Calsyntenin/CASY-1 functions as a kinesin adaptor for Mint/LIN-10 to promote axon degeneration. (**A**) Kinesin binding is required for calsyntenin's function in axon degeneration. Quantification of axon degeneration in 3-day-old adult (3doa). All animals have the *ric-7(n2657); casy-1(wp60)* background. Transgenes and number of animals: none (control) (N = 100); WT CASY-1C (N = 114); CASY-1C with mutated acidic residues (N = 88); CASY-1C with the WW/AA mutation (N = 118). Bars show proportion and 95% CI. ****p<0.0001, ***p<0.001, ns, not significant, compared to

*Figure 4 continued on next page*

*Figure 4 continued*

*ric-7(n2657); casy-1(wp60)* with no transgene except where indicated, Fisher's exact test. Transgenic CASY-1C with the WW/AA mutation does not restore degeneration to *ric-7(n2657); casy-1(wp60)* mutants. (**B**) Loss of kinesin light chain *klc-2* suppresses axon degeneration. Quantification of axon degeneration in 3doa. Genotypes and number of animals: *ric-7(n2657)* (N = 22), *ric-7(n2657), klc-2(km11)* (N = 27), and *ric-7(n2657); klc-1(ok2609)* (N = 20). Bars show proportion and 95% CI. \*\*\*\* p<0.0001, ns, not significant, compared to *ric-7(n2657)*, Fisher's exact test. (**C**) Loss of Mint/*lin-10* suppresses axon degeneration. Quantification of axon degeneration in 3doa. Genotypes and number of animals: *ric-7(n2657)* (N = 50), *ric-7(n2657); lin-10(n1853)* (N = 48), and *ric-7(n2657); lin-10(e1439)* (N = 38). Bars show proportion and 95% CI. \*\*\*\*p<0.0001, compared to *ric-7(n2657)*, Fisher's exact test. (**D**) Images of native and tissue-specific fluorescence (NATF)-tagged calsyntenin/CASY-1 in PVQ cell bodies and distal axons in control, *ric-7(n2657), and klc-2(km11)* animals at early L4 stage. Scale bar, 5 µm. (**E, F**) Loss of kinesin light chain *klc-2* alters calsyntenin abundance. Normalized NATF-tagged CASY-1 intensities (arbitrary units) in PVQ cell bodies and distal axons. Violin plots with median and quantiles are shown. \*\*\* p<0.001, ns, not significant, compared to control, one-way ANOVA, Kruskal–Wallis test, followed by Dunn's multiple comparisons. (**G**) Calsyntenin accumulates in cell body puncta in animals that lack kinesin light chain *klc-2*. Images of NATF-tagged CASY-1 in PVQ cell bodies in control and *klc-2(km11)* at early L4 stage. Excitation and contrast are enhanced in *klc-2(km11)* animals. Scale bar, 5 µm. (**H**) Number of NATF-tagged CASY-1 puncta in PVQ cell bodies. Violin plots with median and quantiles are shown. \*\*\*\*p<0.0001, compared to control, Mann–Whitney test. (**I**) Images of NATF-tagged Mint/LIN-10 in PVQ cell bodies and distal axons in control and *casy-1(wp78)* animals at L4 stage. Boxes highlight the distal axon region for quantifications in (**K**). Scale bar, 5 µm. (**J, K**) Mint/LIN-10 is depleted from distal axon in animals that lack calsyntenin. Normalized NATF-tagged LIN-10 intensities (arbitrary units) in PVQ cell bodies and distal axons. Violin plots with median and quantiles are shown. \*\*\*\*p<0.0001, \*\*p<0.01, compared to control, Mann–Whitney test.

The online version of this article includes the following source data and figure supplement(s) for figure 4:

**Source data 1.** Axon degeneration in CASY-1/KLC/LIN-10 mutants and endogenous CASY-1 and LIN-10 localization.

**Figure supplement 1.** Functional domains in the CASY-1 ICD and quantification of CASY-1C traffic.

**Figure supplement 1—source data 1.** Quantification of CASY-1 vesicle trafficking.

By contrast, loss of KLC-1 did not affect axon degeneration (***Figure 4B***). Together, these data suggest that a kinesin complex containing KLC-2 regulates calsyntenin traffic to promote degeneration.

To test this model, we first examined CASY-1's localization in PVQ axons. We used the NATF approach to label endogenous CASY-1 proteins in PVQ. The *gfp11x7* insertion at the C terminus of the endogenous *casy-1* locus does not interfere with CASY-1's pro-degenerative function as these animals show complete axon degeneration when combined with the *ric-7(n2657)* mutation (***Figure 2—figure supplement 1B***). Endogenous CASY-1 fluorescence is visible in the PVQ cell body and throughout the axon (***Figure 4D***). Loss of *ric-7* does not lead to a significant change in steady-state CASY-1 abundance in the cell body or the distal axon (***Figure 4D–F***). By contrast, loss of KLC-2 results in a dramatic redistribution of CASY-1 to punctate structures located in the cell body and proximal axon (***Figure 4G and H***). This redistribution was accompanied by a near-complete loss of CASY-1 from the distal axon, as well as a partial reduction of CASY-1 levels in the cell body (***Figure 4D–F***). The lack of calsyntenin in the distal axon is consistent with a transport defect, while the reduction in the cell body could result from degradation of accumulated CASY-1 in the soma. Together, these data suggest that CASY-1 localization in axons depends on a kinesin complex containing KLC-2.

To confirm and extend these results, we performed live imaging. The endogenous CASY-1 signals (***Figure 4D***) were too weak for live imaging of trafficking, so we overexpressed CASY-1C with a C terminal GFP tag. This construct preserves CASY-1's pro-degenerative function as its expression in *ric-7(n2657); casy-1(wp78)* animals leads to strong axon degeneration (***Figure 4—figure supplement 1C***). Live imaging of CASY-1C::GFP in *casy-1(wp78)* animals shows that CASY-1 vesicles undergo extensive trafficking in both the proximal and distal regions of the PVQ axon (***Figure 4—figure supplement 1D***). We focused on the distal axon region for analysis. Consistent with the previous finding that calsyntenin interacts with kinesin (***Steuble et al., 2012***), CASY-1C trafficking is highly anterogradely biased (81% of trafficking events are anterograde, ***Figure 4—figure supplement 1E***). For anterograde trafficking, the average speed is 1.07 µm/s, and the average run length is 5.79 µm; for retrograde trafficking, the average speed is 2.08 µm/s, and the average run length is 9.87 µm (***Figure 4—figure supplement 1F and G***). Although we found no change in endogenous *casy-1* distribution in *ric-7(n2657)* animals (***Figure 4D–F***), we did find that CASY-1C trafficking is greatly affected by *ric-7*: the anterograde bias is largely lost (63% are anterograde), and speeds and run lengths in both directions are dramatically reduced (***Figure 4—figure supplement 1F and G***). Because CASY-1 intensity in axon is too weak for trafficking analysis in the *klc-2* mutant, we were unable to directly compare it with the control. Nevertheless, the anterograde bias and speed of calsyntenin traffic (***Figure 4—figure supplement 1D–F***),

together with the dependence of endogenous CASY-1 localization on KLC-2 (*Figure 4D–H*), suggest that trafficking of CASY-1 into distal axons likely depends on kinesin motors.

## Mint/LIN-10 is trafficked by calsyntenin/CASY-1 and is required for degeneration

Calsyntenin/CASY-1 has been shown to function as a kinesin adaptor that transports vesicular and protein cargoes from the neuronal cell body to the distal axon (*Konecna et al., 2006*). For example, mammalian calsyntenin-1 can mediate the anterograde transport of APP (*Ludwig et al., 2009*) and axon guidance receptors such as Robo1 and frizzled 3 (*Alther et al., 2016*). In *C. elegans*, CASY-1 mediates the translocation of a specific insulin receptor isoform, DAF-2C, from the soma to the distal axon in ASE neurons during taste avoidance learning (*Ohno et al., 2014*), and also modulates the transport of synaptic vesicles in GABAergic neurons (*Thapliyal et al., 2018*). It has been proposed that the transport complex consists of four calsyntenin-1 molecules bound to one kinesin-1 motor (*Figure 4—figure supplement 1B*; *Ludwig et al., 2009*). The intracellular region of calsyntenin, in addition to its kinesin interaction, also contains a conserved region that has been shown to bind Mint/ LIN-10 in mammals (*Araki et al., 2003*) and an acidic region that has been proposed to buffer calcium (*Vogt et al., 2001*). To test the importance of the acidic region, we expressed CASY-1C with all the acidic residues in the acidic region mutated to neutral residues (E to Q and D to N). We found that this CASY-1C (mutated acidic) restored strong degeneration to *ric-7(n2657); casy-1(wp60)* animals, but to a lesser extent than the wildtype *casy-1c* transgene (*Figure 4A*). We conclude that the acidic domain is not essential for calsyntenin's pro-degeneration function; the minor reduction in function may be due to a secondary effect on a different function of this region.

To examine the importance of the Mint/LIN-10 interaction, we tested whether Mint/LIN-10 functions similarly to calsyntenin and KLC-2. We found that two independent lof alleles of *lin-10*, *n1853* and *e1439*, suppress axon degeneration (*Figure 4C*), similar to loss of *casy-1* (*Figure 3B*). If LIN-10 is a cargo of CASY-1 vesicles, then LIN-10 localization would be affected in *casy-1* KO animals. We labeled endogenous LIN-10 with the NATF approach. In *casy-1(wp78)* single mutants, LIN-10 abundance in the distal axon is dramatically reduced to 56% of the control level, while its abundance in the soma is increased by 27% (*Figure 4I–K*). In conclusion, our data suggest a model in which CASY-1 transports LIN-10 from the cell body to the distal axon to mediate axon degeneration in the absence of mitochondria. Since CASY-1C has both a TM sequence and KBS, we reasoned that CASY-1C functions as a TM kinesin adaptor that transports Mint/LIN-10 as well as a potential organelle cargo to the distal axon to mediate degeneration (*Figure 4—figure supplement 1B*).

## Loss of calsyntenin/CASY-1 and Mint/LIN-10 suppresses axon degeneration through the CaMKII/UNC-43-Sarm1/TIR-1-ASK1/NSY-1-p38 MAPK pathway

We hypothesized that knocking out calsyntenin/CASY-1 or Mint/LIN-10 protects against degeneration through a specific signaling pathway in axons. It has been shown that calsyntenin, Mint, and APP can form a tripartite complex in mammals (*Araki et al., 2003*), we thus examined the role of the *C. elegans* APP homologue, APL-1, in spontaneous degeneration in *ric-7(n2657)*. We used the Cre-loxP system to knock out the entire *apl-1* locus in PVQ (*Figure 5—figure supplement 1A and B*), but did not observe significant protection against degeneration compared to the P$_{PVQ}$::Cre control (*Figure 5— figure supplement 1C*). We also tested an allele (*wp22*) that deletes most of the APL-1 intracellular region at the C terminus, which has been shown to interact with Mint/LIN-10 (*Tomita et al., 1999*). *ric-7(n2657); apl-1(wp22)* double mutants still show a complete degeneration phenotype (*Figure 5— figure supplement 1C*). Therefore, APL-1 is not involved in the spontaneous degeneration in the absence of mitochondria. LIN-10 has also been shown to form a complex with LIN-2 and LIN-7 to localize EGF receptors in vulval epithelial cells (*Kaech et al., 1998*). However, the *lin-2* lof allele *e1309* does not suppress axon degeneration in *ric-7(n2657)* (*Figure 5—figure supplement 2*), indicating that the LIN-2/7/10 complex (or EGF receptors) is not involved.

Since we identified the *casy-1* and *unc-43* mutations in the same unbiased screen, we next examined the possibility that CaMKII/UNC-43 acts downstream of CASY-1. In *ric-7(n2657); casy-1(wp78)* animals, degeneration is strongly delayed. However, the *unc-43* lof allele *e408* largely abolishes the protection: there is some residual protection in 1doa but degeneration is almost complete in

2doa (*Figure 5A*). *unc-43(e408)* also suppresses *lin-10(n1853)* and *klc-2(km11)* with a similar trend (*Figure 5B and C*). Therefore, the protection against degeneration in *casy-1, lin-10,* and *klc-2* mutants requires the shared downstream factor CaMKII/UNC-43. We could rescue UNC-43 in *ric-7(n2657); casy-1(wp78); unc-43(e408)* triple mutants by expressing WT or CA UNC-43 in PVQ (*Figure 5D*), consistent with the cell-autonomous role of UNC-43 and CASY-1. We next examined if the TIR-1-NSY-1-SEK-1 signaling also functions downstream of CASY-1. Indeed, the lof mutations *tir-1(qd4), nsy-1(ok593), sek-1(km4), pmk-3(ok169),* and *cebp-1(tm2807)* all suppress *casy-1(wp78)* (*Figure 5E*). By contrast, the *pmk-1(km25)* single lof mutation and the *pmk-1(km25); pmk-2(qd279qd171)* double mutation do not suppress *casy-1(wp78)*. Therefore, loss of CASY-1 and LIN-10 acts via the UNC-43-TIR-1-NSY-1-SEK-1-PMK-3-CEBP-1 pathway to protect against axon degeneration in the absence of mitochondria (*Figure 5F*).

## The L-type voltage-gated calcium channel subunits EGL-19 and UNC-36 mediate axon protection in *casy-1* and *lin-10* mutants

Our data suggest that loss of CASY-1 and LIN-10 protect axons by activating CaMKII/UNC-43 and its downstream MAP kinase pathway (*Figure 5F*). How might loss of CASY-1 and LIN-10 increase CaMKII/UNC-43 activity? One possibility is that loss of these factors upregulates CaMKII/UNC-43 abundance in axons in *ric-7(n2657)* mutants. However, endogenous CaMKII/UNC-43 levels in distal PVQ axons are not restored in *ric-7(n2657); casy-1(wp78)* or *ric-7(n2657); lin-10(e1439)* animals compared to the *ric-7(n2657)* control (*Figure 6—figure supplement 1*). Therefore, CASY-1 and LIN-10 do not regulate CaMKII/UNC-43 protein levels in axons. We next tested the hypothesis that loss of CASY-1 or LIN-10 may alter CaMKII/UNC-43 activity by regulating calcium homeostasis. We focused on the VGCCs for two reasons: first, calcium influx through VGCCs can activate CaMKII/UNC-43 (*Sagasti et al., 2001*; *West et al., 2001*); second, Mint/LIN-10 has been shown to directly interact with VGCCs in mammals (*Maximov and Bezprozvanny, 2002*; *Maximov et al., 1999*).

In *C. elegans*, the L-type VGCC $\alpha_1$ subunit, EGL-19, the non-L-type $\alpha_1$ subunit, UNC-2, and the regulatory $\alpha_2\delta$ subunit, UNC-36, have been shown to function upstream of CaMKII/UNC-43 (*Alqadah et al., 2016a*; *Alqadah et al., 2016b*; *Caylor et al., 2013*; *Sagasti et al., 2001*). We thus examined if these subunits are required for protection against axon degeneration in *casy-1* and *lin-10* mutants. We found that the lof mutations *egl-19(n582)* and *unc-36(e251)* completely abolish the delayed degeneration in adult *ric-7(n2657); casy-1(wp78)* animals (*Figure 6A and B*). They also suppress the protection in *ric-7(n2657); lin-10(n1853)* animals, with *ric-7(n2657); lin-10(n1853); egl-19(n582)* triple mutants also displaying enhanced degeneration at L4 (*Figure 6C and D*). By contrast, the lof mutation *unc-2(e55)* does not suppress *casy-1(wp78)* and partially suppresses *lin-10(n1853)* (*Figure 6—figure supplement 2A and B*). We conclude that the L-type VGCC composed of EGL-19 and UNC-36 is required for axon protection in *casy-1* and *lin-10* mutants.

Next, we determined the relationship of the EGL-19/UNC-36 calcium channel to CaMKII/UNC-43 in axon protection. We found that the EGL-19/UNC-36 calcium channel is not required for protection conferred by gain of function in CaMKII/UNC-43: *unc-36(e251); ric-7(n2657); unc-43(n498sd)* animals protected from axon degeneration similar to *ric-7(n2657); unc-43(n498sd)* alone (*Figure 6E*). These data indicate that CaMKII/UNC-43 activity acts downstream of EGL-19/UNC-36 calcium channel, consistent with the model that VGCCs act to activate CaMKII/UNC-43 (*Alqadah et al., 2016a*). Importantly, we found that while EGL-19, UNC-36, and CaMKII/UNC-43 are required to suppress degeneration triggered by loss of mitochondria, loss of any of these factors is not sufficient to trigger axon degeneration in otherwise wild-type animals (*Figure 6—figure supplement 2C*). Thus, the VGCC-CaMKII/UNC-43 pathway is required specifically for axon protection conferred by loss of *casy-1* or *lin-10* in the context of mitochondria mislocalization. Overall, our data suggest a model in which loss of CASY-1 or LIN-10 results in increased neuronal calcium via L-type VGCC. Increased calcium leads to activation of CaMKII/UNC-43. In turn, CaMKII/UNC-43 activity suppresses axon degeneration through the TIR-1-NSY-1-SEK-1 MAPK pathway (*Figure 6F*).

## The calsyntenin-CaMKII-MAPK pathway protects against early-stage degeneration, before axon breakage

Axon degeneration is a sequence of events beginning with axon thinning/beading, followed by axon breakage, fragmentation, and ending with debris clearance (*Figure 1H*). At what step does the

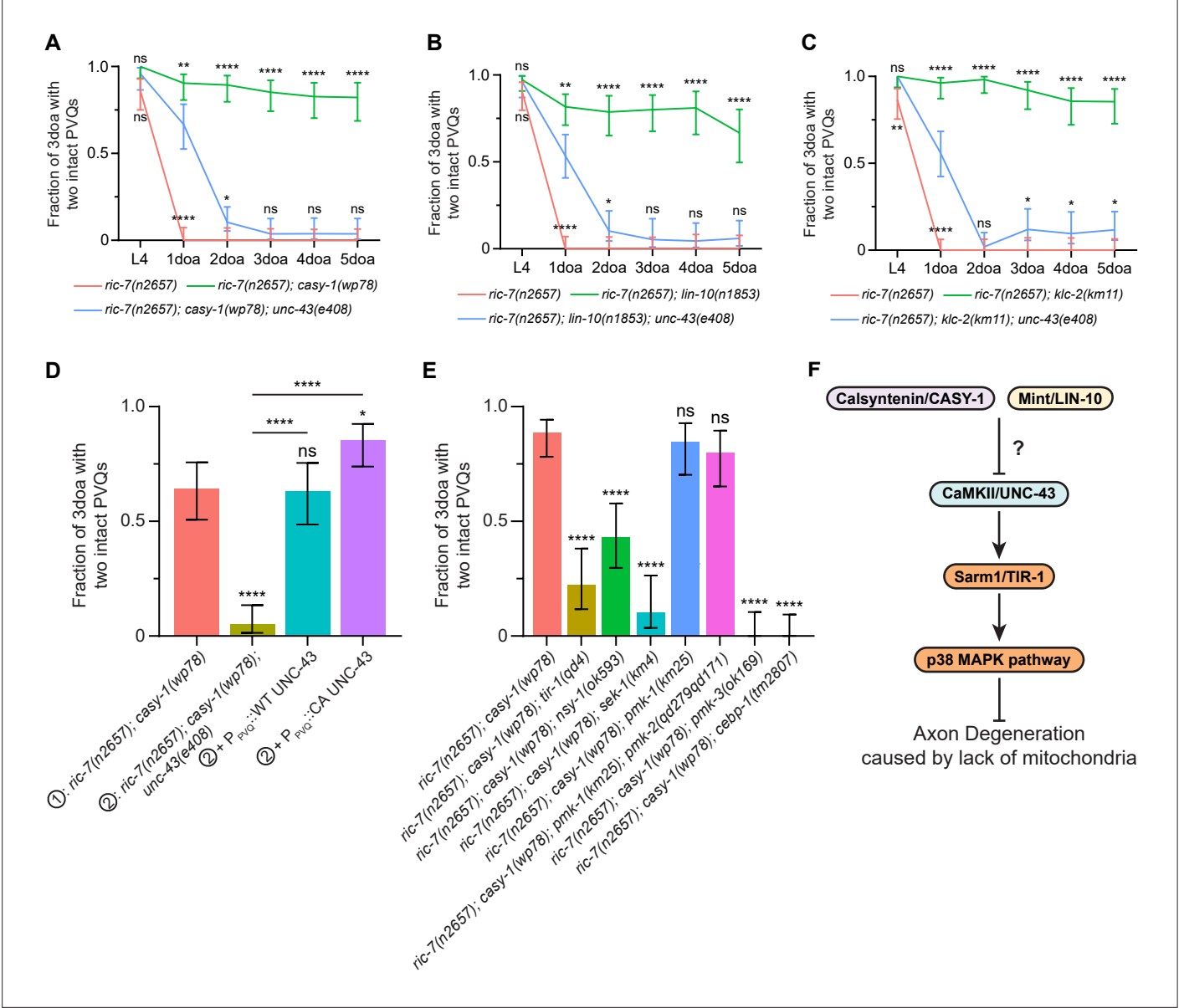

**Figure 5.** Loss of calsyntenin/CASY-1 or Mint/LIN-10 suppresses axon degeneration by activating the CaMKII-Sarm1-ASK1 MAPK pathway. (**A–C**) Axon protection conferred by loss of calsyntenin/*casy-1* requires CaMKII (**A**), Mint (**B**), and kinesin (**C**). Quantification of axon degeneration from L4 to 5-day-old adult (5doa). Number of animals: 45–77 (**A**), 33–75 (**B**), and 42–60 (**C**) animals. Graphs show proportion and 95% CI. **** p<0.0001, **p<0.01, *p<0.05, ns, not significant, compared to the triple mutant (blue curve) at each timepoint, Fisher's exact test. (**D**) CaMKII/*unc-43* is required cell-autonomously to suppress axon degeneration in calsyntenin/*casy-1* mutants. Quantification of axon degeneration in 3-day-old adult (3doa). Genotypes and number of animals: *ric-7(n2657); casy-1(wp78)* (N = 53), *ric-7(n2657); casy-1(wp78); unc-43(e408)* (N = 61), and *ric-7(n2657); casy-1(wp78); unc-43(e408)* + P_PVQ::WT UNC-43 (N = 46) or constitutively active (CA) UNC-43 (N = 55). Bars show proportion and 95% CI. ****p<0.0001, *p<0.05, ns, not significant, compared to *ric-7(n2657); casy-1(wp78)* except where indicated, Fisher's exact test. (**E**) The TIR-1-NSY-1-MAPK pathway is required to suppress axon degeneration in calsyntenin/*casy-1* mutants. Suppression of protection against axon degeneration in *ric-7(n2657); casy-1(wp78)* (N = 61) by *tir-1(qd4)* (N = 36), *nsy-1(ok593)* (N = 44), *sek-1(km4)* (N = 29), *pmk-1(km25)* (N = 39), *pmk-1(km25); pmk-2(qd279qd171)* (N = 40), *pmk-3(ok169)* (N = 33), and *cebp-1(tm2807)* (N = 37). Bars show proportion and 95% CI. ****p<0.0001, ns, not significant, compared to *ric-7(n2657); casy-1(wp78)*, Fisher's exact test. (**F**) Diagram of regulation of axon degeneration by the calsyntenin-Mint-CaMKII-Sarm1-ASK1 p38 MAPK pathway.

The online version of this article includes the following source data and figure supplement(s) for figure 5:

**Source data 1.** Epistatic analysis of mutants in the CASY-1-CaMKII-Sarm1-p38 MAPK pathway.

**Figure supplement 1.** APL-1 mutations do not suppress spontaneous degeneration in *ric-7(n2657)*.

**Figure supplement 1—source data 1.** Gel images of APL-1 genotyping. Three extrachromosomal array lines (*oyIs14[sra-6p::GFP], ric-7(n2657) V; apl-*

*Figure 5 continued on next page*

*Figure 5 continued*

*1(wp19) X + sra-6p::nCre*) were generated from one microinjection and genotyped for APL KO in PVQ. 15 worms from line 1, 8 worms from line 2, and 8 worms from line 3 were genotyped. The KO band (583 bp) can be seen in most worms from line 1 but not in worms from line 2 or line 3, probably due to expression levels and mosaicism. Line 1 was then integrated to generate *XE2634(oyIs14[sra-6p::GFP], ric-7(n2657) V; apl-1(wp19) X; wpIs146[sra-6p::nCre + odr-1p::RFP])*, which was used for analysis of axon degeneration. Control is *XE2415(oyIs14[sra-6p::GFP], ric-7(n2657) V; apl-1(wp19) X)*. Blank is no worm input. The cropped image of control and last three worms from line 1 is shown in Figure 5—figure supplement 1B, with enhanced contrast.

**Figure supplement 1—source data 2.** Axon degeneration in *apl-1* mutants.

**Figure supplement 2.** The *lin-2* lof allele *e1309* does not suppress PVQ degeneration in *ric-7(n2657)*.

**Figure supplement 2—source data 1.** Axon degeneration in the *ric-7; lin-2* mutant.

---

calsyntenin-CaMKII-MAPK pathway prevent degeneration? To test whether the calsyntenin-CaMKII-MAPK pathway can prevent the late steps of degeneration that occur after axon breakage, we used laser axotomy to experimentally induce an axon break in one PVQ axon and examined the degeneration of the distal axon segment 24 hr later. Consistent with previous findings (*Ding and Hammarlund, 2018*; *Nichols et al., 2016*; *Rawson et al., 2014*), injury-induced degeneration of distal PVQ axon segments is slow as nearly all axons are still present 24 hr later (*Figure 1—figure supplement 2A and B*). By contrast, in *ric-7(n2657)* mutants, degeneration after injury is greatly enhanced due to the lack of axonal mitochondria (*Figure 1—figure supplement 2A and B*; *Ding and Hammarlund, 2018*; *Nichols et al., 2016*; *Rawson et al., 2014*). However, loss of calsyntenin or activation of CaMKII did not significantly suppress degeneration in *ric-7(n2657)* after axon transection (*Figure 1—figure supplement 2B*). These data indicate that the calsyntenin-CaMKII-MAPK pathway protects against degeneration by acting before axon breakage and cannot stop degeneration after axon breaks occur. Consistent with this model, protected axons in *ric-7(n2657); casy-1(wp78)* and *ric-7(n2657); unc-43(n498sd)* animals mostly stop at the beading stage and rarely enter the breaking stage (*Figures 2B and 3C*). Therefore, we conclude that the calsyntenin-CaMKII-MAPK pathway specifically prevents the transition from axon thinning and beading towards axon breakage.

## Discussion

Axon degeneration is a critical step in neurodegenerative disease. We used an unbiased approach to identify mechanisms that regulate axon degeneration triggered by loss of axonal mitochondria. We discovered a novel anti-degenerative pathway involving CaMKII/UNC-43, Sarm1/TIR-1, and MAP kinase signaling. In particular, CA CaMKII/UNC-43 activates Sarm1/TIR-1 and the downstream NSY-1-SEK-1-PMK-3-CEBP-1 MAPK pathway to protect against degeneration in a cell-specific manner. Furthermore, we found that the loss of the conserved calsyntenin/Mint/KLC-2 trafficking complex activates the CaMKII-TIR-1-MAPK pathway through the L-type VGCC and suppresses degeneration.

### A key anti-degenerative role for CaMKII

We discovered that CaMKII/UNC-43 is a key factor in regulating axon degeneration when mitochondria are absent. CaMKII is an abundant kinase that is well studied in neurons. Calcium sensitivity is conferred by the interaction with $Ca^{2+}$/calmodulin. Binding of $Ca^{2+}$/calmodulin releases the autoinhibition of CaMKII and activates the enzyme (*Lisman et al., 2002*). CaMKII can also undergo autophosphorylation, leading to persistent activation even after dissociation of calmodulin (*Lisman et al., 2002*). CaMKII in neurons is well studied for its role in long-term potentiation (LTP), which is one of the major mechanisms underlying learning and memory (*Lisman et al., 2002*). $Ca^{2+}$ entry into postsynaptic sites activates CaMKII, which then initiates the signaling cascades that potentiate synaptic transmission (*Lisman et al., 2002*). CaMKII is enriched at postsynaptic density (PSD) and interacts with numerous PSD proteins, including NMDA receptor, AMPA receptor, F-actin, and calcium channels to regulate synaptic function (*Hell, 2014*; *Lisman et al., 2002*; *Zalcman et al., 2018*). Besides its role in learning and memory, neuronal CaMKII has been proposed to function in neurodegeneration, although its role has been unclear. On the one hand, CaMKII has a pro-degenerative function in the context of calcium overload. For example, CaMKII promotes axon degeneration downstream of a mutant TRPV4 channel, where it acts to increase intracellular $Ca^{2+}$ (*Woolums et al., 2020*). On the other hand, in the *Drosophila* retina, failure to activate CaMKII leads to photoreceptor cell death

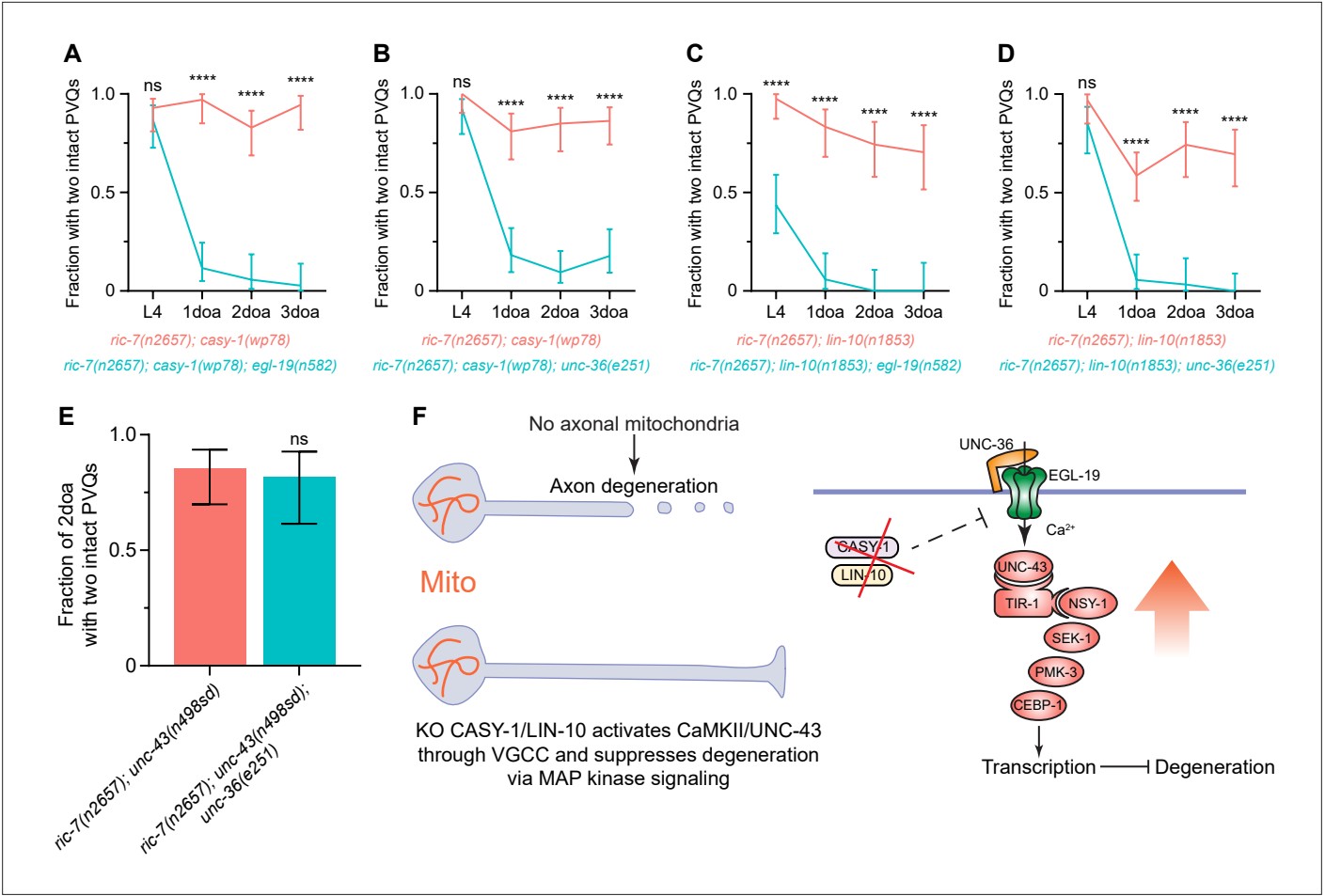

**Figure 6.** Loss of calsyntenin/CASY-1 or Mint/LIN-10 activates CaMKII/UNC-43 through the L-type voltage-gated calcium channel (VGCC) to suppress axon degeneration. (**A–D**) Axon protection conferred by loss of calsyntenin/*casy-1* or Mint/*lin-10* requires VGCC subunit *egl-19* (**A, C**) and VGCC subunit *unc-36* (**B, D**). Quantification of axon degeneration from L4 to 3-day-old adult (3doa). Number of animals = 23–58. Graphs show proportion and 95% CI. ****p<0.0001 at each timepoint, ns, not significant, Fisher's exact test. (**E**) Axon protection conferred by activation of CaMKII does not requires VGCC subunit *unc-36*. Quantification of axon degeneration in 2-day-old adult (2doa). Genotypes and number of animals: *ric-7(n2657); unc-43(n498sd)* (N = 34) and *ric-7(n2657); unc-43(n498sd); unc-36(e251)* (N = 22). Bars show proportion and 95% CI. ns, not significant, Fisher's exact test. (**F**) Integrated model of how loss of calsyntenin/*casy-1* and Mint/*lin-10* suppresses axon degeneration caused by the absence of mitochondria.

The online version of this article includes the following source data and figure supplement(s) for figure 6:

**Source data 1.** Epistatic analysis between *egl-19/unc-36* and *casy-1/lin-10*.

**Figure supplement 1.** CaMKII/UNC-43 abundance in distal axons is not restored in *ric-7(n2657); casy-1(wp78)* and *ric-7(n2657); lin-10(e1439)* animals.

**Figure supplement 1—source data 1.** Endogenous CaMKII abundance in distal axon in control and suppressor mutants.

**Figure supplement 2.** Voltage-gated calcium channel (VGCC) subunit *unc-2* does not mediate axon protection, and loss of CaMKII/*unc-43*, VGCC subunit *egl-19* or VGCC subunit *unc-36* does not result in axon degeneration.

**Figure supplement 2—source data 1.** Epistatic analysis between *unc-2* and *cays-1/lin-10* and degeneration in *unc-43*, *egl-19* and *unc-36* single mutants.

(***Alloway et al., 2000***; ***Kristaponyte et al., 2012***). Consistent with a function in opposing cell death, it was recently shown that activation of CaMKII in mouse retinal ganglion cells protects them against cell death induced by excitotoxicity, optic nerve injury, and glaucoma models (***Guo et al., 2021***). Here, we show that in axon degeneration caused by the absence of mitochondria, CaMKII/UNC-43 acts to oppose axon degeneration: its activation strongly suppresses axon degeneration cell-autonomously.

Our data indicate that CaMKII's effect on axon degeneration is context dependent. Loss of CaMKII/UNC-43 alone does not affect axon stability as the *unc-43(e408)* single mutants do not show spontaneous axon degeneration (***Figure 2A***). However, in the context of axonal mitochondria depletion,

loss of CaMKII accelerates degeneration, while CaMKII activation is neuroprotective. Loss of axonal mitochondria also reduces axonal levels of CaMKII (*Figure 2D and E*). Together, these observations suggest that reduced axonal CaMKII activity contributes to degeneration. However, since even complete loss of CAMKII does not trigger degeneration when axonal mitochondria are normal, loss of axonal mitochondria must trigger parallel pro-degenerative signals besides having effects on CaMKII. This is not surprising since mitochondria perform multifaceted activities to regulate neuronal metabolism. Nonetheless, our work reveals that active CaMKII/UNC-43 can protect against axon degeneration in the absence of mitochondria, which may serve as an entry point for therapeutic interventions.

## CaMKII localization and activation depend on mitochondria

We observed that loss of axonal mitochondria results in reduced axonal CaMKII/UNC-43, suggesting that axonal CaMKII localization or stabilization depends on mitochondria. Precise control of CaMKII localization in neurons is critical for its function. For example, although CaMKII is highly abundant throughout neuronal processes, anchoring at dendritic spines refines its substrate selectivity and is important for LTP (*Hell, 2014*). It has been well studied that CaMKII can translocate to PSD upon stimulation (*Otmakhov et al., 2004*; *Shen and Meyer, 1999*), and the translocation involves interactions with PSD proteins such as NMDA receptors (*Halt et al., 2012*; *Merrill et al., 2005*). However, the mechanisms governing CaMKII transport and abundance in axons are unclear. In *C. elegans*, translocation of CaMKII/UNC-43 to neurites has been shown to be regulated by the interactions between the catalytic and autoinhibitory domain (*Umemura et al., 2005*). Multiple mutations that mimic autophosphorylation result in an increased neurite pool of CaMKII/UNC-43. It has been proposed that unknown transport factors facilitate the trafficking of CaMKII/UNC-43 by binding to its catalytic domain, and conformational changes of CaMKII/UNC-43 affect the interactions with the transport factors (*Umemura et al., 2005*). In this study, we show that axonal mitochondria are important for endogenous CaMKII/UNC-43 localization in axons. It is possible that some CaMKII/UNC-43 transport factors are present on mitochondria, thus facilitating co-trafficking into axons.

In addition to reduced levels of axonal CaMKII/UNC-43, our data indicate that loss of axonal mitochondria also results in deficient CaMKII/UNC-43 activation. Overexpression of WT CaMKII/UNC-43 in *ric-7* animals has only a small protective effect on degeneration (*Figure 2A*). By contrast, mutations that activate endogenous CaMKII, or overexpression of the activated form, strongly suppress degeneration (*Figure 2A*). Thus, CaMKII activation is a key choke point in the ability of axons to withstand degeneration. One possibility is that mitochondria can act as activators of CaMKII/UNC-43 through modulating local calcium homeostasis in axons. Overexpression of calcium-independent CA CaMKII/UNC-43 can thus strongly suppress degeneration, even though axonal mitochondria are absent.

## Kinesin-dependent localization of calsyntenin and Mint controls CaMKII activity and degeneration

We also found that PVQ neurons lacking calsyntenin/CASY-1 are highly resistant to degeneration. Calsyntenins are a family of neuronally enriched TM proteins characterized by the presence of two cadherin-like repeats, an LG/LNS domain in the extracellular region and an intracellular domain (ICD) that carries two KBS (*Araki et al., 2003*; *Konecna et al., 2006*). The three human calsyntenin proteins (CLSTN1–3) regulate diverse neuronal functions. For example, CLSTN1 mediates trafficking of NMDA receptors (*Ster et al., 2014*) and axon guidance receptors (*Alther et al., 2016*), and regulates axon branching (*Ponomareva et al., 2014*) and microtubule polarity (*Lee et al., 2017*). CLSTN2 is involved in learning and memory (*Lipina et al., 2016*; *Preuschhof et al., 2010*). CLSTN3 has been shown to mediate synapse development (*Pettem et al., 2013*). Previous studies have also associated calsyntenins with neurodegeneration, specifically AD pathogenesis (*Gotoh et al., 2020*; *Uchida and Gomi, 2016*; *Vagnoni et al., 2012*). There is also evidence for functional mammalian calsyntenins without the extracellular domain. For example, mammalian calsyntenins undergo proteolytic cleavage by α-secretase to produce the membrane-bound C-terminal fragment (CTF), which can be further cleaved by γ-secretase to produce the cytosolic ICD (*Araki et al., 2004*; *Hata et al., 2009*; *Vogt et al., 2001*). Calsyntenin CTF and ICD have been shown to regulate the trafficking, metabolism, and signaling of amyloid precursor protein (APP) (*Araki et al., 2007*; *Araki et al., 2004*; *Araki et al., 2003*; *Steuble et al., 2012*; *Takei et al., 2015*). In addition, a recent study shows that mouse calsyntenin 3β, which

lacks the extracellular region for cell adhesion, promotes the secretion of S100b from brown adipocytes and stimulates sympathetic innervation (*Zeng et al., 2019*).

We show that in the absence of mitochondria the CASY-1 TM domain and ICD promote degeneration. Specifically, CASY-1's kinesin-binding sites are critical for its pro-degenerative function (*Figure 4A*). Further, loss of *klc-2*, like loss of calsyntenin, suppresses degeneration (*Figure 4B*). The *C. elegans* genome encodes two kinesin light chain genes, *klc-1* and *klc-2*. However, loss of *klc-1* did not phenocopy the *casy-1* or *klc-2* mutant (*Figure 4B*). Consistent with these data, *klc-1* does not appear to be expressed in PVQ (*Taylor et al., 2020*). Further, KLC-1 is a divergent kinesin and its function in axonal transport is unclear (*Sakamoto et al., 2005*), thus, calsyntenin may specifically acts with the kinesin light chain KLC-2 to mediate axon degeneration. We also found that loss of the calsyntenin/CASY-1 cargo, Mint/LIN-10, suppresses degeneration (*Figure 4C*). Therefore, our data support a model in which disruption of a calsyntenin/Mint/KLC-2 trafficking complex prevents axon degeneration in the absence of mitochondria.

## L-type calcium channels link the calsyntenin/Mint/KLC-2 trafficking complex to CaMKII activity and axon protection

We found that axon protection by loss of calsyntenin or Mint requires L-type VGCC. VGCCs are composed of a pore-forming $\alpha_1$ subunit and are usually associated with auxiliary $\beta$ subunits and $\alpha 2\delta$ subunits (*Catterall, 2011*). In *C. elegans*, EGL-19 is the L-type $\alpha_1$ subunit, and UNC-36 is the $\alpha_2\delta$ subunit (*Hobert, 2013*). L-type VGCCs composed of EGL-19 and UNC-36 have critical roles in neurons in calcium influx (*Frøkjaer-Jensen et al., 2006*). However, L-type VGCCs have not been previously shown to mediate neurodegeneration. We found that loss of *egl-19* or *unc-36* completely blocks the protective effect of disrupting the calsyntenin/Mint/KLC-2 trafficking complex (*Figure 6A–D*). As L-type channels are known to bind Mint, these data suggest that disrupting the calsyntenin/Mint/KLC-2 trafficking complex changes the localization or organization of L-type calcium channels (*Figure 6F*). In turn, reorganized calcium channels protect axons.

What is the target of reorganized calcium channels? In *C. elegans*, EGL-19 and UNC-36 have been shown to activate CaMKII/UNC-43 (*Alqadah et al., 2016a*; *Alqadah et al., 2016b*; *Caylor et al., 2013*; *Sagasti et al., 2001*). In mammals, L-type channels can act as CaMKII anchor proteins at postsynaptic sites and allow efficient CaMKII stimulation upon $Ca^{2+}$ influx (*Abiria and Colbran, 2010*; *Hudmon et al., 2005*; *Rose et al., 2009*). We found that L-type calcium channels are not required for axon protection by activated CaMKII (*Figure 6E*). These data suggest that a key result of reorganized calcium channels is the activation of CaMKII. If CaMKII is activated by a gain-of-function mutation, its activity is not dependent on calcium and does not require reorganized L-type channels.

How and where does CASY-1 and LIN-10 activate the CaMKII-TIR-1-MAPK pathway through the L-type VGCC? We speculate that CASY-1/LIN-10 can regulate VGCC localization or activity. For example, CASY-1 and LIN-10 may mediate the export of VGCCs from the soma and its delivery into the distal axon. Therefore, in CASY-1, LIN-10, or KLC-2 KO, VGCCs may accumulate in the soma, leading to more $Ca^{2+}$ entry and hyperactivation of CaMKII. Eventually, active CEBP-1 enters the nucleus to regulate transcription, which in turn suppresses degeneration. Alternatively, CASY-1 and LIN-10 that are transported into distal axons may inhibit VGCC activity, leading to less $Ca^{2+}$ entry and reduced CaMKII activity. Therefore, in *casy-1* or *lin-10* mutants, axonal CaMKII is hyperactive and activates the TIR-1-MAPK pathway. Active PMK-3 or CEBP-1 may then be retrogradely transported back to the soma to instruct transcriptional changes. In favor of the second model, LIN-10, CaMKII, TIR-1, and NSY-1 have been shown to colocalize at synaptic regions in distal axons (*Chuang and Bargmann, 2005*), and L-type VGCCs are enriched at postsynaptic sites (*Hell et al., 1996*). Further experiments that examine CaMKII/UNC-43 and MAPK activity in distinct compartments will help disentangle these two models.

## The surprising neuroprotective role of the Sarm1/TIR-1-ASK1/NSY-1-p38 MAPK pathway

We discovered a surprising axon-protective role of the *C. elegans* Sarm1 homologue, TIR-1 (CeTIR-1). Sarm1 was initially identified as an essential activator of injury-induced axon degeneration in flies and mammals (*Gerdts et al., 2013*; *Osterloh et al., 2012*). Sarm1 promotes axon degeneration by regulating $NAD^+$ metabolism (*Essuman et al., 2017*; *Gerdts et al., 2015*). Specifically, Sarm1 proteins

form octamers through the sterile alpha motifs (SAM) domains (*Horsefield et al., 2019*; *Sporny et al., 2019*). The Toll/interleukin-1 receptor homology (TIR) domain possesses intrinsic capabilities to hydrolyze $NAD^+$, but its activity is normally autoinhibited by the Armadillo/HEAT (ARM) repeat domain in healthy axons (*Essuman et al., 2017*). Upon axon injury, the autoinhibition is released, causing rapid $NAD^+$ collapse, which eventually leads to axon degeneration. Sarm1 was later shown to mediate other types of axon degeneration such as in traumatic brain injury (*Henninger et al., 2016*) and peripheral neuropathy (*Turkiew et al., 2017*). Interestingly, there is evidence for Sarm1 involvement in neurodegeneration downstream of drug-induced mitochondria dysfunction (*Loreto et al., 2020*; *Summers et al., 2014*; *Summers et al., 2019*). However, other forms of neurodegeneration, such as SOD1-ALS, do not require Sarm1 (*Peters et al., 2018*), indicating that multiple pro-degeneration mechanisms may exist. *C. elegans* Sarm1 is well conserved with its counterparts in other species, containing the ARM domains, two SAM domains, and a TIR domain. Further, activated *C. elegans* TIR domains have been shown to cause $NAD^+$ depletion (*Horsefield et al., 2019*) and trigger cell death in cultured mouse neurons (*Summers et al., 2016*). Therefore, CeTIR-1 likely possesses the capacity to deplete $NAD^+$ and promote degeneration, albeit less efficiently than its human and fly counterparts (*Horsefield et al., 2019*; *Summers et al., 2016*). However, so far, there is little evidence that the $NAD^+$ hydrolase activity of CeTIR-1 is essential to axon degeneration in *C. elegans* in vivo. Loss of CeTIR-1 does not protect against axon degeneration induced by laser axotomy in *C. elegans* (*Nichols et al., 2016*). Overexpressing the murine $Wld^S$ or the *C. elegans* Nmnats, which promote $NAD^+$ synthesis, does not delay axon degeneration (*Nichols et al., 2016*). Our data show that loss of CeTIR-1 or overexpressing Nmnats does not suppress spontaneous axon degeneration caused by the loss of mitochondria. Together, these data suggest that $NAD^+$ consumption by CeTIR-1 does not play an essential role in promoting axon degeneration in *C. elegans*. However, given its intimate relationship with degeneration, CeTIR-1 may regulate degeneration through different or additional signaling in *C. elegans*.

Quite unexpectedly, we found that Sarm/TIR-1 is required for the protection against axon degeneration by active UNC-43/CaMKII, and this protection is mediated by the conserved NSY-1-SEK-1 MAPK pathway. In general, Sarm/TIR-1 family members have a complex relationship with MAP kinase signaling. Sarm1 has been shown to coordinate with different MAP kinase components to promote axon degeneration in mammals. Specifically, the MAPKKKs MEKK4, MLK2, and DLK, the MAPKKs MKK4 and MKK7, and the MAPKs JNK1-3 are required for axon degeneration after injury (*Walker et al., 2017*; *Yang et al., 2015*). However, an in vivo study in *Drosophila* did not observe the involvement of JNK in injury-induced axon degeneration (*Neukomm et al., 2017*). The NSY-1-SEK-1 pathway is a different MAPK cascade: NSY-1(MAPKKK) is the homologue of human ASK1(or MEKK5), SEK-1(MAPKK) is the homologue of human MKK3/6, and the downstream MAPKs are PMKs, which fall in the p38 rather than the JNK family (*Hayakawa et al., 2006*). The discrepancy between the neuroprotective role that we observed here and the previously identified pro-degenerative role of the TIR-1-MAPK pathways may be due to the specificity of the MAPK pathways that are activated in these different contexts. It is possible that distinct MAPK pathways are activated during different types of degeneration and function through distinct downstream factors. It has been well established that the TIR-1-NSY-1-SEK-1-PMK-1 signaling acts in innate immunity to resist bacterial pathogens (*Kim et al., 2002*). The pathway also functions in the nervous system to regulate asymmetric neural development, with PMK-1 and PMK-2 acting redundantly (*Alqadah et al., 2016b*; *Pagano et al., 2015*). Interestingly, a previous study has shown that this pathway promotes motor neuron degeneration in a *C. elegans* model of ALS (*Vérièpe et al., 2015*). In these animals, secreted molecules from neurons that overexpress human TDP-43 or FUS activate the transcription factor ATF-7 through the TIR-1-MAPK pathway and lead to neurodegeneration. Here, PMK-1 and ATF-7 form a linear pathway downstream of SEK-1 to mediate degeneration. By contrast, in our *ric-7(n2657)* model where degeneration is caused by the lack of mitochondria, PMK-3 and the transcription factor CEBP-1 confer protection downstream of TIR-1, NSY-1 ,and SEK-1, which highlights that the role of MAPK signaling is highly context dependent in neurodegeneration. Whether the protective role of the CaMKII-TIR-1-p38 MAPK axis is conserved in other species and other degeneration models awaits further investigation.

## An active mechanism that overrides neurodegeneration

Mitochondria defects are tightly associated with neurodegeneration. Because lack of axonal mitochondria results in completely penetrant PVQ axon degeneration, this model offers an opportunity to discover cellular mechanisms that control degeneration. This work identifies a linear CaMKII/UNC-43-Sarm1/TIR-1-ASK1/NSY-1 MAPK pathway as an active neuroprotective program that suppresses axon degeneration due to loss of axonal mitochondria. The surprising protective role of these molecules in this context highlights the complex nature of axon degeneration. In contexts where axon degeneration is associated with defects in mitochondria location or function, the calsyntenin-CaMKII-MAPK pathway may provide intervention points for suppressing degeneration.

# Materials and methods

**Key resources table**

| Reagent type (species) or resource | Designation | Source or reference | Identifiers | Additional information |
|---|---|---|---|---|
| Chemical compound, drug | Levamisole hydrochloride | Santa Cruz Biotechnology | Cat# sc-205730 | |
| Chemical compound, drug | Proteinase K | Sigma | Cat# 3115879001 | |
| Other | Polybead Microspheres 0.05 μm | Polysciences | Cat# 08691 | |
| Commercial assay, kit | Gateway LR Clonase II Enzyme mix | Invitrogen | Cat# 11791020 | |
| Commercial assay, kit | 1 kb DNA Ladder | Promega | Cat# G571A | |
| Peptide, recombinant protein | Alt-R S.p. Cas9 Nuclease V3 | IDT | Cat# 1081058 | |
| RNA | tracrRNA | IDT | Cat# 1072532 | |
| Chemical compound, drug | Ethyl methanesulfonate | Sigma | Cat# M0880 | |
| Commercial assay, kit | QIAprep Spin Miniprep Kit | QIAGEN | Cat# 27106 | |
| Peptide, recombinant protein | Phusion High-Fidelity DNA Polymerase | NEB | Cat# M0530L | |
| Chemical compound, drug | Nonidet P-40 | americanBIO | Cat# AB01425 | |
| Chemical compound, drug | TWEEN 20 | Sigma | Cat# P5927 | |
| Chemical compound, drug | Gelatin | MP Biomedicals | Cat# 901771 | |
| Chemical compound, drug | β-Mercaptoethanol | Sigma | Cat# M3148-25ML | |
| Chemical compound, drug | UltraPure phenol:chloroform:isoamyl alcohol | Invitrogen | Cat# 15593031 | |
| Chemical compound, drug | Sodium acetate, 3 M solution, pH5.2 | americanBIO | Cat# AB13168-01000 | |
| Chemical compound, drug | Nuclease-free water | americanBIO | Cat# AB02123-00500 | |
| Genetic reagent (*Caenorhabditis elegans* N2 hermaphrodite) | *oyIs14[sra-6p::GFP] V* | CGC | XE2047 | |
| Genetic reagent (*C. elegans* N2 hermaphrodite) | *oyIs14[sra-6p::GFP], ric-7(n2657) V* | This study | XE2046 | Control; Generated in the Hammarlund lab |

*Continued on next page*

*Continued*

| Reagent type (species) or resource | Designation | Source or reference | Identifiers | Additional information |
|---|---|---|---|---|
| Genetic reagent (*C. elegans* N2 hermaphrodite) | *oyIs14[sra-6p::GFP] V; wpEx369[sra-6p::mito::TagRFP+ odr-1p::RFP]* | This study | XE2263 | Mito marker; Generated in the Hammarlund lab |
| Genetic reagent (*C. elegans* N2 hermaphrodite) | *oyIs14[sra-6p::GFP], ric-7(n2657) V; wpEx369[sra-6p::mito::TagRFP+ odr-1p::RFP]* | This study | XE2264 | Mito marker: Generated in the Hammarlund lab |
| Genetic reagent (*C. elegans* N2 hermaphrodite) | *oyIs14[sra-6p::GFP], ric-7(n2657) V; casy-1(wp60) II* | This study | XE2209 | *casy-1 lof* suppressor; Generated in the Hammarlund lab |
| Genetic reagent (*C. elegans* N2 hermaphrodite) | *oyIs14[sra-6p::GFP], ric-7(n2657) V; unc-43(wp64) II* | This study | XE2210 | *unc-43 gof* suppressor; Generated in the Hammarlund lab |
| Genetic reagent (*C. elegans* N2 hermaphrodite) | *ric-7(n2657) V; casy-1(wp78) II; wpEx370[sra-6p::casy-1c::GFP+ sra-6p::TagRFP+ odr-1p::RFP]* | This study | XE2265 | *casy-1* OE; Generated in the Hammarlund lab |
| Genetic reagent (*C. elegans* N2 hermaphrodite) | *casy-1(wp78) II; wpEx370[sra-6p::casy-1c::GFP+ sra-6p::TagRFP+ odr-1p::RFP]* | This study | XE2266 | *casy-1* OE; Generated in the Hammarlund lab |
| Genetic reagent (*C. elegans* N2 hermaphrodite) | *oyIs14[sra-6p::GFP], ric-7(n2657) V; unc-43(wp64) IV; wpEx369[sra-6p::mito::TagRFP+ odr-1p::RFP]* | This study | XE2618 | Mito marker; Generated in the Hammarlund lab |
| Genetic reagent (*C. elegans* N2 hermaphrodite) | *oyIs14[sra-6p::GFP], ric-7(n2657) V; casy-1(wp60) II; wpEx371[sra-6p::mito::TagRFP+ odr-1p::RFP]* | This study | XE2269 | Mito marker; Generated in the Hammarlund lab |
| Genetic reagent (*C. elegans* N2 hermaphrodite) | *oyIs14[sra-6p::GFP], ric-7(n2657) V; casy-1(wp60) II; wpEx374[sra-6p::casy-1a::SL2::mcherry+ myo-2p::mcherry]* | This study | XE2274 | *casy-1* OE; Generated in the Hammarlund lab |
| Genetic reagent (*C. elegans* N2 hermaphrodite) | *oyIs14[sra-6p::GFP], ric-7(n2657) V; casy-1(wp60) II; wpEx375[sra-6p::casy-1b::SL2::mcherry+ myo-2p::mcherry]* | This study | XE2275 | *casy-1* OE; Generated in the Hammarlund lab |
| Genetic reagent (*C. elegans* N2 hermaphrodite) | *oyIs14[sra-6p::GFP], ric-7(n2657) V; casy-1(wp60) II; wpEx376[sra-6p::casy-1c::SL2::mcherry+ myo-2p::mcherry]* | This study | XE2276 | *casy-1* OE; Generated in the Hammarlund lab |
| Genetic reagent (*C. elegans* N2 hermaphrodite) | *oyIs14[sra-6p::GFP], ric-7(n2657) V; casy-1(wp60) II; wpEx379[sra-6p::casy-1a no KBS::SL2::mcherry+ myo-2p::mcherry]* | This study | XE2277 | *casy-1* OE; Generated in the Hammarlund lab |
| Genetic reagent (*C. elegans* N2 hermaphrodite) | *oyIs14[sra::gfp], ric-7(n2657) V; casy-1(wp60) II; wpEx407[sra-6p::casy-1c_delta_extracellular::SL2::mCherry+ myo-2p::mCherry];* | This study | XE2355 | *casy-1* OE; Generated in the Hammarlund lab |
| Genetic reagent (*C. elegans* N2 hermaphrodite) | *oyIs14[sra-6p::GFP], ric-7(n2657) V; casy-1(wp60) II; wpEx380[sra-6p::casy-1 cytoplasmic domain::SL2::mcherry+ myo-2p::mcherry]* | This study | XE2278 | *casy-1* OE; Generated in the Hammarlund lab |
| Genetic reagent (*C. elegans* N2 hermaphrodite) | *oyIs14[sra-6p::GFP] V; casy-1(wp60) II* | This study | XE2374 | *casy-1(wp60)* suppressor; Generated in the Hammarlund lab |
| Genetic reagent (*C. elegans* N2 hermaphrodite) | *oyIs14[sra-6p::GFP] V; casy-1(wp60) II; wpEx376[sra-6p::casy-1c::SL2::mcherry+ myo-2p::mcherry]* | This study | XE2544 | *casy-1* OE; Generated in the Hammarlund lab |
| Genetic reagent (*C. elegans* N2 hermaphrodite) | *oyIs14[sra-6p::GFP], ric-7(n2657) V; wpEx438[sra-6p::casy-1c mutated acidic region::SL2::mcherry+ myo-2p::mcherry]* | This study | XE2547 | *casy-1* OE; Generated in the Hammarlund lab |

*Continued on next page*

*Continued*

| Reagent type (species) or resource | Designation | Source or reference | Identifiers | Additional information |
|---|---|---|---|---|
| Genetic reagent (*C. elegans* N2 hermaphrodite) | *oyIs14[sra::gfp], ric-7(n2657) V; casy-1(wp60) II; wpEx431[sra-6p::casy-1(WW-AA)::SL2::mCherry+ myo-2p::mCherry]* | This study | XE2504 | *casy-1* OE; Generated in the Hammarlund lab |
| Genetic reagent (*C. elegans* N2 hermaphrodite) | *oyIs14[sra-6p::GFP], ric-7(n2657) V; wpEx381[sra-6p::unc-116::GFP::tomm-7+ sra-6p::mito::TagRFP+ odr-1p::RFP]* | This study | XE2279 | Kinesin-mito chimera; Generated in the Hammarlund lab |
| Genetic reagent (*C. elegans* N2 hermaphrodite) | *oyIs14[sra-6p::GFP], ric-7(n2657), klc-2(km11) V* | This study | XE2350 | *klc-2* suppressor; Generated in the Hammarlund lab |
| Genetic reagent (*C. elegans* N2 hermaphrodite) | *oyIs14[sra-6p::GFP], ric-7(n2657) V; klc-1(ok2609) IV* | This study | XE2290 | *klc-1* mutant; Generated in the Hammarlund lab |
| Genetic reagent (*C. elegans* N2 hermaphrodite) | *oyIs14[sra-6p::GFP], ric-7(n2657) V; tir-1(qd4) III* | This study | XE2308 | *tir-1 lof*; Generated in the Hammarlund lab |
| Genetic reagent (*C. elegans* N2 hermaphrodite) | *oyIs14[sra-6p::GFP], ric-7(n2657) V; tir-1(ok2859) III* | This study | XE2309 | *tir-1 lof*; Generated in the Hammarlund lab |
| Genetic reagent (*C. elegans* N2 hermaphrodite) | *oyIs14[sra-6p::GFP], ric-7(n2657) V; casy-1(tm718) II* | This study | XE2294 | *casy-1(tm718)*; Generated in the Hammarlund lab |
| Genetic reagent (*C. elegans* N2 hermaphrodite) | *oyIs14[sra-6p::GFP], ric-7(n2657) V; casy-1(wp78) II* | This study | XE2262 | *casy-1(wp78)* suppressor; Generated in the Hammarlund lab |
| Genetic reagent (*C. elegans* N2 hermaphrodite) | *casy-1(wp78) II* | This study | XE2260 | *casy-1* KO; Generated in the Hammarlund lab |
| Genetic reagent (*C. elegans* N2 hermaphrodite) | *oyIs14[sra-6p::GFP], ric-7(n2657) V; casy-1(wp60); wpEx397[WRM0622dH03+ myo-2p::mCherry]* | This study | XE2313 | *casy-1* fosmid rescue; Generated in the Hammarlund lab |
| Genetic reagent (*C. elegans* N2 hermaphrodite) | *oyIs14[sra-6p::GFP], ric-7(n2657) V; unc-43(n498sd) IV* | This study | XE2419 | *unc-43(n498sd)* suppressor; Generated in the Hammarlund lab |
| Genetic reagent (*C. elegans* N2 hermaphrodite) | *oyIs14[sra-6p::GFP], ric-7(n2657) V; unc-43(e408) IV* | This study | XE2423 | *unc-43 lof*; Generated in the Hammarlund lab |
| Genetic reagent (*C. elegans* N2 hermaphrodite) | *ric-7(n2657) V; casy-1(wp78) II* | This study | XE2244 | *casy-1* KO; Generated in the Hammarlund lab |
| Genetic reagent (*C. elegans* N2 hermaphrodite) | *oyIs14[sra-6p::GFP], ric-7(n2657) V; lin-10(n1853) I* | This study | XE2428 | *lin-10* suppressor; Generated in the Hammarlund lab |
| Genetic reagent (*C. elegans* N2 hermaphrodite) | *oyIs14[sra-6p::GFP], ric-7(n2657) V; unc-43(e408) IV; casy-1(wp78) II* | This study | XE2429 | *unc-43 lof*; *casy-1*; Generated in the Hammarlund lab |
| Genetic reagent (*C. elegans* N2 hermaphrodite) | *wpIs141[sra-6p::GFP1−10+ myo-2p::mcherry]* | This study | XE2441 | PVQ::GFP1-10; Generated in the Hammarlund lab |
| Genetic reagent (*C. elegans* N2 hermaphrodite) | *oyIs14[sra-6p::GFP], ric-7(n2657) V; lin-10(n1853) I; unc-43(e408) IV* | This study | XE2508 | *unc-43 lof*; *lin-10*; Generated in the Hammarlund lab |

*Continued*

| Reagent type (species) or resource | Designation | Source or reference | Identifiers | Additional information |
|---|---|---|---|---|
| Genetic reagent (*C. elegans* N2 hermaphrodite) | *oyIs14[sra-6p::GFP], ric-7(n2657) V; lin-10(e1439) I* | This study | XE2490 | *lin-10* suppressor; Generated in the Hammarlund lab |
| Genetic reagent (*C. elegans* N2 hermaphrodite) | *oyIs14[sra-6p::GFP] V; unc-43(e408) IV* | This study | XE2580 | *unc-43 lof*; Generated in the Hammarlund lab |
| Genetic reagent (*C. elegans* N2 hermaphrodite) | *oyIs14[sra-6p::GFP], ric-7(n2657) V; unc-43(e408) IV; casy-1(wp78) II + wpEx430[sra-6p::unc-43g + myo-2p::mcherry]* | This study | XE2496 | WT *unc-43* OE; Generated in the Hammarlund lab |
| Genetic reagent (*C. elegans* N2 hermaphrodite) | *oyIs14[sra-6p::GFP], ric-7(n2657) V; unc-43(e408) IV; casy-1(wp78) II + wpEx432[sra-6p::unc-43g T284D::SL2::mcherry+ myo-2p::mcherry]* | This study | XE2506 | Constitutively active *unc-43* OE; Generated in the Hammarlund lab |
| Genetic reagent (*C. elegans* N2 hermaphrodite) | *oyIs14[sra-6p::GFP], ric-7(n2657) V; lin-10(e1439) I; unc-43(e408) IV* | This study | XE2530 | *unc-43 lof; lin-10*; Generated in the Hammarlund lab |
| Genetic reagent (*C. elegans* N2 hermaphrodite) | *oyIs14[sra-6p::GFP], ric-7(n2657) V; unc-2(e55) X; casy-1(wp78) II* | This study | XE2546 | *unc-2; casy-1*; Generated in the Hammarlund lab |
| Genetic reagent (*C. elegans* N2 hermaphrodite) | *ric-7(n2657) V; unc-43(wp106[unc-43::gfp11x7]) IV; wpIs141[sra-6p::GFP1−10+ myo-2p::mcherry] II* | This study | XE2564 | Endogenous UNC-43; Generated in the Hammarlund lab |
| Genetic reagent (*C. elegans* N2 hermaphrodite) | *ric-7(n2657) V; unc-43(wp106[unc-43::gfp11x7]) IV; wpIs141[sra-6p::GFP1−10+ myo-2p::mcherry], casy-1(wp78) II* | This study | XE2565 | Endogenous UNC-43; Generated in the Hammarlund lab |
| Genetic reagent (*C. elegans* N2 hermaphrodite) | *unc-43(wp106[unc-43::gfp11x7]) IV; wpIs141[sra-6p::GFP1−10+ myo-2p::mcherry] II* | This study | XE2567 | Endogenous UNC-43; Generated in the Hammarlund lab |
| Genetic reagent (*C. elegans* N2 hermaphrodite) | *ric-7(n2657) V; lin-10(e1439) I; unc-43(wp106[unc-43::gfp11x7]) IV; wpIs141[sra-6p::GFP1−10+ myo-2p::mcherry] II* | This study | XE2605 | Endogenous UNC-43; Generated in the Hammarlund lab |
| Genetic reagent (*C. elegans* N2 hermaphrodite) | *ric-7(n2657) V; unc-43(wp106[unc-43::gfp11x7]) IV; wpIs141[sra-6p::GFP1−10+ myo-2p::mcherry] II; wpEx450[sra-6p::unc-116::tomm-7+ odr-1p::RFP]* | This study | XE2649 | Endogenous UNC-43 with forced transport of mito; Generated in the Hammarlund lab |
| Genetic reagent (*C. elegans* N2 hermaphrodite) | *casy-1(syb3311[casy-1::gfp11x7]) II* | This study | PHX3311 | *casy-1::gfp11x7* KI; Generated by SunyBiotech |
| Genetic reagent (*C. elegans* N2 hermaphrodite) | *oyIs14[sra-6p::GFP], ric-7(n2657), klc-2(km11) V; unc-43(e408)* | This study | XE2575 | *unc-43 lof; klc-2*; Generated in the Hammarlund lab |
| Genetic reagent (*C. elegans* N2 hermaphrodite) | *casy-1(syb3311[casy-1::gfp11x7]), wpIs141[sra-6p::GFP1−10+ myo-2p::mcherry] II* | This study | XE2593 | Endogenous *casy-1*; Generated in the Hammarlund lab |
| Genetic reagent (*C. elegans* N2 hermaphrodite) | *casy-1(syb3311[casy-1::gfp11x7]), wpIs141[sra-6p::GFP1−10+ myo-2p::mcherry] II; ric-7(n2657) V* | This study | XE2594 | Endogenous CASY-1; Generated in the Hammarlund lab |
| Genetic reagent (*C. elegans* N2 hermaphrodite) | *casy-1(syb3311[casy-1::gfp11x7]), wpIs141[sra-6p::GFP1−10+ myo-2p::mcherry] II; klc-2(km11) V* | This study | XE2595 | Endogenous CASY-1; Generated in the Hammarlund lab |
| Genetic reagent (*C. elegans* N2 hermaphrodite) | *oyIs14[sra-6p::GFP], ric-7(n2657) V; unc-2(e55) X; lin-10(n1853) I* | This study | XE2585 | *unc-2 lof; lin-10*; Generated in the Hammarlund lab |

*Continued on next page*

*Continued*

| Reagent type (species) or resource | Designation | Source or reference | Identifiers | Additional information |
|---|---|---|---|---|
| Genetic reagent (*C. elegans* N2 hermaphrodite) | *lin-10(wp109[lin-10::gfp11x7]) I; wpIs141[sra-6p::GFP1–10+ myo-2p::mcherry] II* | This study | XE2586 | Endogenous LIN-10; Generated in the Hammarlund lab |
| Genetic reagent (*C. elegans* N2 hermaphrodite) | *lin-10(wp109[lin-10::gfp11x7]) I; wpIs141[sra-6p::GFP1–10+ myo-2p::mcherry] casy-1(wp78) II* | This study | XE2587 | Endogenous LIN-10; Generated in the Hammarlund lab |
| Genetic reagent (*C. elegans* N2 hermaphrodite) | *oyIs14[sra-6p::GFP], ric-7(n2657) V; egl-19(n582) IV; casy-1(wp78) II* | This study | XE2606 | *egl-19 lof; casy-1*; Generated in the Hammarlund lab |
| Genetic reagent (*C. elegans* N2 hermaphrodite) | *oyIs14[sra-6p::GFP], ric-7(n2657) V; egl-19(n582) IV; lin-10(n1853) I* | This study | XE2607 | *egl-19 lof; lin-10*; Generated in the Hammarlund lab |
| Genetic reagent (*C. elegans* N2 hermaphrodite) | *oyIs14[sra-6p::GFP], ric-7(n2657) V; unc-36(e251) III; casy-1(wp78) II* | This study | XE2608 | *unc-36 lof; casy-1*; Generated in the Hammarlund lab |
| Genetic reagent (*C. elegans* N2 hermaphrodite) | *oyIs14[sra-6p::GFP], ric-7(n2657) V; unc-36(e251) III; lin-10(n1853) I* | This study | XE2609 | *unc-36 lof; lin-10*; Generated in the Hammarlund lab |
| Genetic reagent (*C. elegans* N2 hermaphrodite) | *oyIs14[sra-6p::GFP], ric-7(n2657) V; casy-1(syb3311[casy-1::gfp11x7]) II* | This study | XE2610 | *casy-1 gfp11x7* KI; Generated in the Hammarlund lab |
| Genetic reagent (*C. elegans* N2 hermaphrodite) | *oyIs14[sra-6p::GFP], ric-7(n2657) V; apl-1(wp19) X* | This study | XE2415 | *apl-1(wp19)*; Generated in the Hammarlund lab |
| Genetic reagent (*C. elegans* N2 hermaphrodite) | *oyIs14[sra-6p::GFP], ric-7(n2657) V; apl-1(wp19) X; wpIs146[sra-6p::nCre+ odr-1p::RFP]* | This study | XE2634 | *apl-1* KO; Generated in the Hammarlund lab |
| Genetic reagent (*C. elegans* N2 hermaphrodite) | *oyIs14[sra-6p::GFP], ric-7(n2657) V; wpIs146[sra-6p::nCre+ odr-1p::RFP]* | This study | XE2698 | Cre control; Generated in the Hammarlund lab |
| Genetic reagent (*C. elegans* N2 hermaphrodite) | *oyIs14[sra-6p::GFP], ric-7(n2657) V; apl-1(wp22) X* | This study | XE2489 | *apl-1(wp22)*; Generated in the Hammarlund lab |
| Genetic reagent (*C. elegans* N2 hermaphrodite) | *oyIs14[sra-6p::GFP] V; egl-19(n582) IV* | This study | XE2640 | *egl-19 lof*; Generated in the Hammarlund lab |
| Genetic reagent (*C. elegans* N2 hermaphrodite) | *oyIs14[sra-6p::GFP] V; unc-36(e251) III* | This study | XE2641 | *unc-36 lof*; Generated in the Hammarlund lab |
| Genetic reagent (*C. elegans* N2 hermaphrodite) | *oyIs14[sra-6p::GFP], ric-7(n2657) V; unc-43(n498sd) IV; unc-36(e251) III* | This study | XE2648 | *unc-36 lof; unc-43 gof*; Generated in the Hammarlund lab |
| Genetic reagent (*C. elegans* N2 hermaphrodite) | *oyIs14[sra-6p::GFP], ric-7(n2657) V; unc-43(n498sd) IV; tir-1(qd4) III* | This study | XE2652 | *tir-1 lof; unc-43 gof*; Generated in the Hammarlund lab |
| Genetic reagent (*C. elegans* N2 hermaphrodite) | *oyIs14[sra-6p::GFP], ric-7(n2657) V; unc-43(n498sd) IV; nsy-1(ok593) II* | This study | XE2653 | *nsy-1 lof; unc-43 gof*; Generated in the Hammarlund lab |
| Genetic reagent (*C. elegans* N2 hermaphrodite) | *oyIs14[sra-6p::GFP], ric-7(n2657) V; unc-43(n498sd), pmk1(km25) IV* | This study | XE2655 | *pmk-1 lof; unc-43 gof*; Generated in the Hammarlund lab |

*Continued*

| Reagent type (species) or resource | Designation | Source or reference | Identifiers | Additional information |
|---|---|---|---|---|
| Genetic reagent (*C. elegans* N2 hermaphrodite) | oyIs14[sra-6p::GFP], ric-7(n2657) V; casy-1(wp78) II; tir-1(qd4) III | This study | XE2656 | *tir-1 lof; casy-1* KO; Generated in the Hammarlund lab |
| Genetic reagent (*C. elegans* N2 hermaphrodite) | oyIs14[sra-6p::GFP], ric-7(n2657) V; casy-1(wp78), nsy-1(ok593) II | This study | XE2657 | *nsy-1 lof; casy-1* KO; Generated in the Hammarlund lab |
| Genetic reagent (*C. elegans* N2 hermaphrodite) | oyIs14[sra-6p::GFP], ric-7(n2657) V; casy-1(wp78) II; sek-1(km4) X | This study | XE2658 | *sek-1 lof; casy-1* KO; Generated in the Hammarlund lab |
| Genetic reagent (*C. elegans* N2 hermaphrodite) | oyIs14[sra-6p::GFP], ric-7(n2657) V; casy-1(wp78) II; pmk-1(km25) IV | This study | XE2659 | *pmk-3 lof; casy-1* KO; Generated in the Hammarlund lab |
| Genetic reagent (*C. elegans* N2 hermaphrodite) | oyIs14[sra-6p::GFP], ric-7(n2657) V; wpEx453[sra-6p::unc-43g T284D::SL2::mcherry+ myo-2p::mcherry] | This study | XE2661 | Constitutively active *unc-43* OE; Generated in the Hammarlund lab |
| Genetic reagent (*C. elegans* N2 hermaphrodite) | oyIs14[sra-6p::GFP], ric-7(n2657) V; wpEx452[sra-6p::unc-43g + myo-2p::mcherry] | This study | XE2662 | Constitutively active *unc-43* OE; Generated in the Hammarlund lab |
| Genetic reagent (*C. elegans* N2 hermaphrodite) | oyIs14[sra-6p::GFP] V; unc-116(rh24sb79) III | This study | XE2411 | *unc-116 lof*; Generated in the Hammarlund lab |
| Genetic reagent (*C. elegans* N2 hermaphrodite) | oyIs14[sra-6p::GFP] V; unc-116(rh24sb79) III; unc-43(n498sd) IV | This study | XE2717 | *unc-116 lof; unc-43 gof*; Generated in the Hammarlund lab |
| Genetic reagent (*C. elegans* N2 hermaphrodite) | ric-7(n2657) V; unc-43(wp106[unc-43::gfp11x7]) IV; wpEx475[sra-6p::unc-116::GFP1−10+ myo-2p::mCherry] | This study | XE2752 | Endogenous UNC-43 with forced transport of mitochondria; Generated in the Hammarlund lab |
| Genetic reagent (*C. elegans* N2 hermaphrodite) | oyIs14[sra-6p::GFP], ric-7(n2657) V; unc-43(wp106[unc-43::gfp11x7]) IV; wpEx475[sra-6p::unc-116::GFP1−10+ myo-2p::mCherry] | This study | XE2753 | Endogenous UNC-43 with forced transport of mitochondria; Generated in the Hammarlund lab |
| Genetic reagent (*C. elegans* N2 hermaphrodite) | oyIs14[sra-6p::GFP] V; miro-1(wy50180) IV; mtx-2(wy50266) III | This study | XE2839 | *miro-1 lof; mtx-2 lof*; Generated in the Hammarlund lab |
| Genetic reagent (*C. elegans* N2 hermaphrodite) | oyIs14[sra-6p::GFP] V; trak-1(gk571211) I | This study | XE2840 | *trak-1 lof*; Generated in the Hammarlund lab |
| Genetic reagent (*C. elegans* N2 hermaphrodite) | oyIs14[sra-6p::GFP] V; miro-1(tm1966) IV; miro-2(tm2933) X | This study | XE2349 | *miro-1 lof; miro-2 lof*; Generated in the Hammarlund lab |
| Genetic reagent (*C. elegans* N2 hermaphrodite) | oyIs14[sra-6p::GFP] V; mtx-2(wy50266) III; miro-1(wy50180) IV; casy-1(wp78) II | This study | XE2840 | *miro-1 lof; mtx-2 lof; casy-1* KO; Generated in the Hammarlund lab |
| Genetic reagent (*C. elegans* N2 hermaphrodite) | oyIs14[sra-6p::GFP] V; mtx-2(wy50266) III; miro-1(wy50180), unc-43(n498sd) IV; | This study | XE2878 | *miro-1 lof; mtx-2 lof; unc-43 gof*; Generated in the Hammarlund lab |
| Genetic reagent (*C. elegans* N2 hermaphrodite) | oyIs14[sra-6p::GFP], ric-7(n2657) V; wpEx499[sra-6p::nmat-1::SL2::mCherry+ myo-2p::mCherry] | This study | XE2910 | *nmat-1 OE*; Generated in the Hammarlund lab |

*Continued*

| Reagent type (species) or resource | Designation | Source or reference | Identifiers | Additional information |
|---|---|---|---|---|
| Genetic reagent (*C. elegans* N2 hermaphrodite) | *oyIs14[sra-6p::GFP], ric-7(n2657) V; wpEx500[sra-6p::nmat-2::SL2::mCherry+ myo-2p::mCherry]* | This study | XE2911 | *nmat-2* OE; Generated in the Hammarlund lab |
| Genetic reagent (*C. elegans* N2 hermaphrodite) | *oyIs14[sra-6p::GFP] V; mtx-2(wy50266) III; miro-1(wy50180) IV; wpEx369[sra-6p::mito::TagRFP+ odr-1p::RFP]* | This study | XE2883 | *miro-1 lof; mtx-2 lof* with mito marker; Generated in the Hammarlund lab |
| Genetic reagent (*C. elegans* N2 hermaphrodite) | *oyIs14[sra-6p::GFP] V; miro-1(tm1966) IV; miro-2(tm2933) X; wpEx501[sra-6p::mito::TagRFP+ myo-2p::mCherry]* | This study | XE2912 | *miro-1 lof; miro-2 lof* with mito marker; Generated in the Hammarlund lab |
| Recombinant DNA reagent | sra-6p::casy-1a CDS::SL2::mCherry | This study | pCD1 | PVQ::casy-1a; Generated in the Hammarlund lab |
| Recombinant DNA reagent | sra-6p::casy-1b CDS::SL2::mCherry | This study | pCD2 | PVQ::casy-1b; Generated in the Hammarlund lab |
| Recombinant DNA reagent | sra-6p::casy-1c CDS::SL2::mCherry | This study | pCD3 | PVQ::casy-1c; Generated in the Hammarlund lab |
| Recombinant DNA reagent | sra-6p::casy-1a no KBS CDS::SL2::mCherry | This study | pCD4 | PVQ::casy-1a no KBS; Generated in the Hammarlund lab |
| Recombinant DNA reagent | sra-6p::casy-1 cytoplasmic CDS::SL2::mCherry | This study | pCD5 | PVQ::casy-1 cytoplasmic domain; Generated in the Hammarlund lab |
| Recombinant DNA reagent | sra-6p::casy-1 CDS no ECD::SL2::mCherry | This study | pYW135 | PVQ::casy-1 with out ECD; Generated in the Hammarlund lab |
| Recombinant DNA reagent | sra-6p::mito::tagRFP | This study | pCD6 | PVQ::mito marker; Generated in the Hammarlund lab |
| Recombinant DNA reagent | sra-6p::unc-116::gfp::tomm7 | This study | pCD7 | PVQ::mito-gfp-chimera; Generated in the Hammarlund lab |
| Recombinant DNA reagent | sra-6p::unc-43g WT | This study | pCD8 | PVQ::WT unc-43 isoform g; Generated in the Hammarlund lab |
| Recombinant DNA reagent | sra-6p::unc-43g T286D::SL2::mCherry | This study | pCD9 | PVQ::CA unc-43; Generated in the Hammarlund lab |
| Recombinant DNA reagent | sra-6p::GFP1-10 | This study | pCD10 | PVQ::GFP1-10; Generated in the Hammarlund lab |
| Recombinant DNA reagent | sra-6p::unc-116::tomm-7 | This study | pCD11 | PVQ::mito-chimera; Generated in the Hammarlund lab |
| Recombinant DNA reagent | sra-6p::casy-1c CDS-WWAA::SL2::mCherry | This study | pCD12 | PVQ::casy-1c with the WW/AA mutation; Generated in the Hammarlund lab |
| Recombinant DNA reagent | sra-6p::casy-1c CDS::mutated acidic region::SL2::mCherry | This study | pYW184 | PVQ::casy-1c with mutated acidic residues; Generated in the Hammarlund lab |

*Continued on next page*

*Continued*

| Reagent type (species) or resource | Designation | Source or reference | Identifiers | Additional information |
|---|---|---|---|---|
| Recombinant DNA reagent | sra-6p::casy-1c::GFP | This study | pYW86 | PVQ::casy-1c::GFP; Generated in the Hammarlund lab |
| Recombinant DNA reagent | sra-6p::Cre | This study | pCD13 | PVQ::Cre; Generated in the Hammarlund lab |
| Recombinant DNA reagent | pRF4::rol-6(su1006) | *Mello et al., 1991* | pRF4 | |
| Recombinant DNA reagent | sra-6p::wt tir-1a::SL2::mCherry | This study | pCD14 | PVQ::WT tir-1 isoform a; Generated in the Hammarlund lab |
| Recombinant DNA reagent | sra-6p::unc-116::gfp1-10 | This study | pCD15 | PVQ::kinesin-gfp1-10; Generated in the Hammarlund lab |
| Recombinant DNA reagent | sra-6p::nmat-1::SL2::mCherry | This study | pCD27 | PVQ::nmat-1; Generated in the Hammarlund lab |
| Recombinant DNA reagent | sra-6p::nmat-2::SL2::mCherry | This study | pCD28 | PVQ::nmat-2: Generated in the Hammarlund lab |
| Software, algorithm | Fiji 1.53c | NIH | https://imagej.nih.gov/ij/download.html | |
| Software, algorithm | MetaMorph, version 7.10.2.240 | Molecular Devices | https://mdc.custhelp.com/app/products/detail/p/13 | |
| Software, algorithm | Prism, version 8 | GraphPad | https://www.graphpad.com/scientific-software/prism/ | |
| Software, algorithm | Galaxy | *Afgan et al., 2016* | https://usegalaxy.org/ | |
| Software, algorithm | MiModD | *Maier and Baumeiste, 2016* | http://mimodd.readthedocs.io/en/latest/index.html | |
| Software, algorithm | Bowtie2-2.3.4.2 | *Langmead and Salzberg, 2012* | http://bowtie-bio.sourceforge.net/bowtie2/index.shtml | |
| Software, algorithm | ApE | M.Wayne Davis | https://jorgensen.biology.utah.edu/wayned/ape/ | |
| Software, algorithm | Jalview, version 2.11.1.2 | *Waterhouse et al., 2009* | http://www.jalview.org/getdown/release/ | |

## C. elegans

Worms were maintained at 20°C on NGM plates seeded with OP50 *Escherichia coli*. Hermaphrodites were used for all the assays in this study. Worm strains in this study are listed in the Key resources table.

## Generation of transgenic worms and CRISPR

Transgenic animals were generated by microinjection with Promega 1 kb DNA ladder as a filler in the injection mix. Plasmids for injection were assembled using Gateway recombination (Invitrogen). Entry clones for Gateway reactions were generated by Gibson Assembly (*Gibson et al., 2009*).

CRISPR/Cas9 was used to generate deletions and knock-ins according to a recent improved protocol (*Ghanta and Mello, 2020*). Briefly, 0.5 µl of 10 µg/ul Cas9 protein was mixed with 5 µl of 0.4 µg/ul tracrRNA and 2.8 µl of 0.4 µg/µl crRNA from IDT and was incubated @ 37°C for 15 min. Then 25 ng/

**Table 1.** List of key oligonucleotides for CRISPR.

| Name | Sequence | Source |
|---|---|---|
| **crRNA(20 bp immediately 5' to the PAM sequence)** | | |
| *casy-1(wp78)* 5' | gtgagggtggaaaatgattg | IDT |
| *casy-1(wp78)* 3' | tgtgatgtaatcaacagggt | IDT |
| *unc-43-gfp11x7* | gagaaaaataggcataaaga | IDT |
| *casy-1-gfp11x7#1* | agaacgagcgttcgttgaga | SunyBiotech |
| *casy-1-gfp11x7#2* | tgtcgttggaggtcttgagt | SunyBiotech |
| *lin-10-gfp11x7* | ataaacaatcaaatgtattg | IDT |
| **Genotyping primers** | | |
| *casy-1(wp78)* | f: gaataagaatgagaagacccgctgc; r1: ctccttgcagattgattattggcgc; r2: aaggagtgaaaaggacagtatgaagacg | IDT |
| *unc-43-gfp11x7* | f: tcagaaacggagaagctcatacccg; r1: tcatgtagtaccatatggtcgcgtcc; r2: ccatatatctgagagaatgggacaag | IDT |
| *casy-1-gfp11x7* | f: aaattccttcaggcatgttg; r: gaaggagtgaaaaggacagt | IDT |
| *lin-10-gfp11x7* | f: tcgcagttgcacatgacaggtgag; r: attcacattagggcgcactttctgg | IDT |
| **Others** | | |
| *7XGFP11* template | tcaggaggccgtgaccacatggtccttcatgagtatgtaaa tgctgctgggattacaggtggctctggaggtagagatcatat ggttctccacgaatacgttaacgccgcaggcatcactggcgg tagtggaggacgcgcaccatatggtactacatgaatatgtcaatg cagccggaataaccggagggtccggaggccgggatcacat ggtgctgcatgagtatgtgaacgcggcgggtataactggtggg tcgggcggacgagaccatatggtgcttcacgaatacgtaaacg cagctggcattactggcggatcaggtggcagggatcacatggt actccatgagtacgtgaacgctgctggaatcacaggcggtagcgg cggtcgggaccatatggtcctgcacgaatatgtcaatgctg ccggtatcaccggcggcaag | IDT |
| *GFP1-10* template | atgtccaaaggagaagaactgtttacgggtgttgtgccaatttt ggttgaactcgatggtgatgtcaacggacataagttctcagtg agaggcgaaggagaaggtgacgccaccattggaaaattg actcttaaattcatctgtactactggtaaacttcctgtaccatgg ccgactctcgtaacaacgcttacgtacggagttcagtgcttttc gagatacccagaccatatgaaaagacatgacttttttaagtc ggctatgcctgaaggttacgtgcaagaaagaacaatttcgttc aaagatgatggaaaatataaaactagagcagttgttaaattt gaaggagatactttggttaaccgcattgaactgaaaggaacag attttaaagaagatggtaatattcttggacacaaactcgaatacaa ttttaatagtcataacgtatacatcactgctgataagcaaaagaac ggaattaaagcgaatttcacagtacgccataatgtagaagatgg cagtgttcaacttgccgaccattaccaacaaaacacccctattgg agacggtccggtacttcttcctgataatcactacctctcaacacaaa cagtcctgagcaaagatccaaatgaaaaataa | IDT |

µl double-stranded DNA donor with 40–50 bp homology arms was added to the mixture. 1.6 µl of 500 ng/µl PRF4::*rol-6(su1006)* plasmid was also added as a selection marker (***Mello et al., 1991***). Nuclease-free water was added to bring the final volume to 20 µl. The mixture was then centrifuged @ 15,000 rpm for 2 min. 16 µl of the mixture was transferred to a fresh tube and was kept on ice before injection. To increase the frequency of homology-directed repair, the dsDNA donor amplified by PCR was melted and cooled before use according to the protocol (***Ghanta and Mello, 2020***). F1 non-rollers were then selected for genotyping to find heterozygous mutants. Homozygotes were sequenced to confirm the edits. crRNAs, genotyping primers, and repair templates can be found in ***Table 1***.

The *casy-1::gfp11x7* KI strain PHX3311 was generated by SunyBiotech using CRISPR/Cas9 with two guide RNAs simultaneously (see ***Table 1***).

## Fosmid preparation

The bacteria clone that contains the WRM0622dH03 fosmid was grown overnight in 3 ml LB with 1.1 µl chloramphenicol. 6 µl 500X CopyControl Fosmid Autoinduction Solution was also added to induce the fosmid to high copy according to the CopyControl Fosmid Library Production Kits (Epicentre). The fosmid was then prepared using the standard QIAGEN miniprep kit.

## Quantification of axon degeneration

Worms at the desired age were immobilized with 10 mM levamisole in M9 and mounted on a 3% agarose pad on a glass slide. Worms were then examined under a Zeiss Axioplan 2 microscope equipped with a Zeiss Plan-APOCHROMAT 63×/1.4 oil objective and a pE-300 LED from CoolLED. Fractions of worms with two nontruncated (intact) PVQ axons were quantified.

## *ric-7* suppressor screen and mapping

*XE2046(oyIs14[sra-6p::GFP], ric-7(n2657) V)* worms were mutated with 47 mM EMS. Three F1s were placed on each plate. We screened around 50 F2 adults from each plate using a Zeiss Axioplan 2 microscope with a Zeiss 63×/1.4 oil objective. A total of 340 plates were screened. Animals with strong suppression of PVQ truncation were recovered. The suppression phenotype was further confirmed in the following generations.

Mutants from the screen were backcrossed with the parent strain *XE2046(oyIs14[sra-6p::GFP], ric-7(n2657) V)* 3–4 times. Genomic DNA were extracted using phenol/CHCl$_3$ extraction from the backcrossed worms. Briefly, 10 plates of freshly starved worms were collected with M9, washed, and centrifuged to obtain ~100 µl pellet. The pellet was then lysed with 350 µl lysis buffer (50 mM KCl, 10 mM Tris pH 8.3, 25 mM MgCl$_2$, 0.45% Nonidet P-40, 0.45% Tween-20%, and 0.01% gelatin), 12 µl 20% SDS, 2 µl 20 mg/ml proteinase K, and 1 µl β-mercaptoethanol at 65°C (water bath) for 2 hr. Next, 235 µl lysis buffer was added to bring the final volume to 700 µl. 700 µl of phenol/CHCl$_3$ was then added and mixed with the lysate. The solution was left resting for 1 min until both phases were defined. The tube was centrifuged for 5 min at 15,000 rpm at room temperature. 500 µl supernatant was transferred to a new tube and another round of phenol/CHCl3 was performed to get 400 µl supernatant. Ethanol/NaOAc precipitation was then performed by adding 50 µl 3 M NaOAc (1:10) and 1250 µl 100% ice-cold ethanol (2.5 volumes) and incubating on ice for 10 min. Precipitated nucleic acids and salts were spun down at 4°C, and the supernatant was removed. 400 µl EB buffer was added, and the tube was incubated at 37°C (water bath) for 10 min to dissolve the DNA. 2.3 µl P1 buffer from the QIAGEN miniprep kit that contains RNAse was then added, and the tube was incubated at 37°C (water bath) for 30 min to remove RNA. Another round of phenol/CHCl$_3$ extraction was performed to get 400 µl supernatant. Then ethanol/NaOAc precipitation was done to obtain the DNA pellet. The pellet was washed with 70% ice-cold ethanol three times to remove salts. The pellet was then air dried for 10 min and dissolved in 50 µl nuclease-free water at 37°C overnight. The DNA concentration was then measured and examined by electrophoresis the next day. The final genomic DNA were sent for whole-genome sequencing at the Yale Center for Genome Analysis. The parent strain was also sequenced as the control. Sequencing was performed using NovaSeq S4 paired-end 150 bp with each sample taking up 1.5% of a lane. The final coverage was around 70–150×.

Sequencing results were analyzed using MiModD (*Maier and Baumeiste, 2016*) on the galaxy server (*Afgan et al., 2016*). Briefly, reads were aligned to the reference genome (the WS220/ce10 assembly) using Bowtie2 (*Langmead and Salzberg, 2012*). Homozygous variants in each mutant strain were then extracted based on the same reference genome using mimodd_varcall and mimodd_varextract. Variants were filtered using mimodd_vcf_filter so that only those that are present in one mutant strain but not in other mutant strains or the parent strain were kept. The filtered variants were then rebased to the ce11 assembly using mimodd_rebase and annotated with the SnpEff4.3 WBcel235.86 file. The annotated variants were finally exported using the MiModD Report Variants tool.

Usually, the variants in a certain mutant strain are enriched on one chromosome since backcrosses removed most unlinked variants. The mutations predicted to have the most severe effect on protein coding (stop gain, splicing mutations) on that chromosome were then further tested by fosmid rescue or mutants.

## Imaging

Worms were immobilized with 10 mM levamisole and mounted on a 3% agarose pad on a glass slide. To image the entire PVQ neuron including the nerve ring, we used a spinning disc confocal microscope (Nikon Ti-E Eclipse inverted scope equipped with 488 and 561 laser lines, a Prior NZ250 Piezo stage, a PerkinElmer UltraVIEW VoX, a Hamamatsu C9100-50 camera, and a CFI Plan Fluor 40× oil objective [1.3 NA]). Volocity software (PerkinElmer) was used to acquire images and automatically stitch them. Images that only cover a part of PVQ (no stitching) were performed on a Leica DMi8 microscope equipped with a Visitech i-SIM super-resolution confocal system, 488, 561, and 637 laser lines, an ASI-XYpZ Piezo stage and a Hamamatsu ORCA-Flash4.0 CMOS camera. MetaMorph Imaging software (Molecular Devices) was used to acquire these images.

Imaging analysis was performed with ImageJ. Image z-stacks were first projected using maximum projection. To quantify mitochondria density, mitochondria puncta in axons were identified based on manual thresholding. Both the proximal axon region and the distal axon region in each worm were included. For endogenous protein labeling, GFP intensity was background-subtracted, measured, and normalized to the control. In the distal axon, we focused on the region before the axon joins the nerve ring and becomes undistinguishable from the head neurons. PVQ axons shown in figures are often straightened in ImageJ.

Time-course images of PVQ degeneration from L4 to 1doa were acquired on the i-SIM microscope. Individual worms were first imaged at early L4 stage (0 hr) as described above. Worms were then transferred to OP50 seeded plates and allowed to recover before they were imaged again. Each worm was imaged five times in total (0 hr, 4 hr, 8 hr, 12 hr, and 24 hr).

## Laser axotomy

Laser axotomy was performed as described previously (*Byrne et al., 2011*). L4 animals were immobilized with M9 containing 0.05 µm diameter polystyrene beads (Polysciences) and mounted on a 3% agarose pad on a glass slide. Animals were visualized with a Nikon Eclipse 80i microscope using a 100× Plan Apo VC lens (1.4 NA). One of the two PVQ axons in each worm was cut at the vulval region with 10 pulses of a 435 nm Micropoint laser at 20 Hz. Animals were then recovered to OP50-seeded NGM plates and analyzed 24 hr later for axon degeneration.

## Calsyntenins sequence alignment

Alignment of calsyntenin protein sequences was performed in Jalview (*Waterhouse et al., 2009*) using the ClustalO algorithm with default settings.

## Quantification and statistical analysis

For axon degeneration analyses, fraction of worms with two intact (nontruncated) axons was quantified, and the data were expressed as proportion ± 95% confidence of interval. Two-sided Fisher's exact tests were performed on these binary data in GraphPad Prism. For nonbinary data, multiple conditions were compared by Kruskal–Wallis nonparametric ANOVA test, followed by Dunn's multiple comparisons test. Other statistical tests are also described in the figure legends.

## Materials availability

Plasmids and strains utilized in this study are listed in the Key resources table and will be made available to the scientific community upon request directed to Dr. Hammarlund. Strains generated in this study will be deposited at the Caenorhabditis Genetics Center (CGC).

# Acknowledgements

We thank Daniel Colon-Ramos's lab for sharing fosmids. We thank Shaul Yogev's lab for sharing strains, discussion of the project, and guidance on whole-genome sequencing. We thank the Yale Center for Genome Analysis for performing the whole-genome sequencing. We thank the Cellular Neuroscience, Neurodegeneration and Repair (CNNR) imaging core for microscopy. We thank the international Caenorhabditis Genetics Center for strains. We are grateful to members of the Hammarlund lab for valuable comments.

## Additional information

### Funding

| Funder | Grant reference number | Author |
|---|---|---|
| Chan Zuckerberg Initiative | | Marc Hammarlund |
| Kavli Foundation | | Marc Hammarlund |
| National Institutes of Health | R01 NS098817 | Marc Hammarlund |
| National Institutes of Health | R01 NS094219 | Marc Hammarlund |

The funders had no role in study design, data collection and interpretation, or the decision to submit the work for publication.

### Author contributions

Chen Ding, Conceptualization, Formal analysis, Investigation, Methodology, Visualization, Writing - original draft, Writing - review and editing; Youjun Wu, Hadas Dabas, Formal analysis, Investigation, Writing - review and editing; Marc Hammarlund, Conceptualization, Funding acquisition, Resources, Supervision, Visualization, Writing - original draft, Writing - review and editing

### Author ORCIDs

Chen Ding http://orcid.org/0000-0002-1054-9668
Hadas Dabas http://orcid.org/0000-0002-6654-0873
Marc Hammarlund http://orcid.org/0000-0002-3068-068X

### Decision letter and Author response

Decision letter https://doi.org/10.7554/eLife.73557.sa1
Author response https://doi.org/10.7554/eLife.73557.sa2

## Additional files

### Supplementary files

• Transparent reporting form

### Data availability

Figure 5-figure supplement 1-source data 1 contains the original gel image used to generate Figure 5-figure supplement 1B. All other data generated or analysed during this study are included in the manuscript.

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
