## [Editor Report]

This work reports exciting findings that show that activation of CaMKII downstream of the L-type Ca channel protects axons from RIC-7-induced axon degeneration. The authors also report the surprising result that the worm SARM, TIR-1 together with NSY-1 and SEK-1, mediates this protection effect.

---

## [Decision Letter]

**Decision letter after peer review:**

Thank you for submitting your article "Activation of the CaMKII-Sarm1-ASK1 MAP kinase pathway protects against axon degeneration caused by loss of mitochondria" for consideration by *eLife*. Your article has been reviewed by 3 peer reviewers, including Kang Shen as Reviewing Editor and Reviewer #1, and the evaluation has been overseen by Piali Sengupta as the Senior Editor. The following individual involved in review of your submission has agreed to reveal their identity: Marc Freeman (Reviewer #3).

The reviewers have discussed their reviews with one another, and the Reviewing Editor has drafted this to help you prepare a revised submission. All reviewers agreed that this is a thorough genetic analyses of the mechanisms of a specific type of axonal degeneration induced by mitochondria defects. We all think that the results about the SARM/tir-1's potential protective function is unexpected and can be potentially interesting. The reviewers did raise a number technical questions as below. However, the only essential revision is to revise the text and abstract to make sure that you do not overstate the conclusions. Please make it clear that the mechanisms apply to this specific form of axon degeneration.

Essential revisions:

1) Please follow reviewer 3's suggestion to revise text.

2) Please consider using the suggestions made by all three reviewers in preparing your revised manuscript.

*Reviewer #1:*

Chen and colleagues made full advantage of the power of genetic to uncover novel pathways in axon degeneration and protection. They started with the ric-7 mutant which showed strong lack of mitochondria in *C. elegans* axons. They previously showed that ric-7 causes axonal degeneration. Using forward genetic screens, the discovered that mutations in CaMKII or casy-1 protects the axons from degenerating. Through extensive genetic analyses, they showed that the activation of CaMKII downstream of the L-type Ca channel protect axon from ric-7 induced axon degeneration. Surprisingly, the worm SARM, TIR-1 together with NSY-1 and SEK-1 mediates this protection effect. This is unexpected because SARM has been shown in both flies and mammals to promote axon degeneration.

The genetics experiments are of high quality. The suppression or desuppression phenotypes are very dramatic. The authors have also extensively used endogenously label proteins to study their localization, which avoid the commonly encountered overexpression problems. Overall, this is an exciting manuscript and deserves to be published in *eLife*.

Of course, because of the number of molecules involved in this pathway, there will be new things to be learned in the future. Much of my technical comments include some of the directions that I believe will be interesting. I don't see these comments as necessary to be addressed for this current manuscript except point No.1 and 2.

1. To what extent, does miro-1 miro-2 double mutant still have axonal mitochondria in PVQ axon? This will help us to understand to what extent mitochondria have to be lost in order to trigger axonal degeneration.

2. Since there are many phenotypes in kinesin 1 or klc-2 mutants, the changes of casy-1 distribution in the klc-2 is not the strongest argument for direct transport of casy-1 by kinesin motor complex. The conclusions should reflect this possibility. It would be interesting to look at casy-1 trafficking in the klc-2 mutant. I feel that "demonstrate that *C. elegans* calsyntenin is anterogradely trafficked by kinesin, line 31" is too strong.

3. Is there an explanation for why the trafficking parameters of casy-1 is altered in the ric-7 mutant?

4. How does LIN-10 affect ca^2+^ channel localization? LIN-10, LIN-2 and LIN-7 forms a complex to localize EGF receptors. Are LIN-2 and LIN-7 involved in degeneration protection.

5. How does mitochondria distribution defect alter the Ca dynamics in the axon?

*Reviewer #2:*

7 years ago, the fact the suppressing the ability of mitochondria to traffic down axons leads to increased "fragility" and tendency to degenerate was established. The work presented here sought to understand why loss of mitochondria would do this. The authors used classical suppressor genetic screening to identify molecular elements necessary for this induced fragile state. The screen worked: the authors found several important new pieces of information building strong evidence for an "anti-degenerative" pathway. The findings are novel, robust and unexpected and they have identified several linked pieces. Given the interest in general in understanding the pathways of axon-degeneration, this view of 'anti-degeneration" will likely be impactful on the field.

This is therefore an interesting and important paper. It describes many new facets of axon degeneration with the remarkable observation that Sarm can be prodegenerative when mitochondria are not present, and important and surprising twist. The story is nicely complimented by the connection to the CASY mutation, through likely activation of egl19 leading to activation of CamKII (unc43).

I have very few quibbles with this work

1) I think the authors should be more cautious in how they interpret the issue of unc-43 localization. All enzymes need to be localized to find their substrates. The fact that ric-7 leads to a restriction of un-43 to the cell body is interesting but the fact that moving some of it to the distal axon did not rescue the degeneration I don't think speaks very deeply to whether localization matters. The fact is we don't know where un-43 needs to be, and that forcing WT un-43 to the end of the axon does not rescue things is more or less a failed experiment. A more interesting experiment would be to see how specific localizations of WT versus active unc-43 impacts degeneration.

2) Related to the point above: perhaps having too much WT unc-43 in the cell body is the real problem and the active mutant simply does a better job of leaving the cell body.

The issues above serve as examples of why caution should be exercised in interpreting the localization experiments.

3) The EGL19 link is a potentially satisfying way to link up the unc43 hit with the CASY hit. A compelling experiment would be to use crisper/Cas9 to modify the channel conductance so that it could no longer serve as the potential source of Ca to activate unc43.

*Reviewer #3:*

The authors sought to understand how loss of ric-7, and in turn mitochondria from axons, leads to axon degeneration. In a forward genetic screen they identify CamkII and Casy, which led to their genetic analysis of CamkII/tir-1/NSY1 and cay/mint/kin in axon degeneration. The genetic analyses are excellent, clear and well done. Loss of either of these pathway appears to suppress ric-7-induced axon degeneration. In general, the authors claims are supported by their data, although I think they need to take care in extending their claims to other systems (see below). The potential impact of the work on the field is not entirely clear to me. The major surprise would be that tir-1 (SARM1 in mammals) plays a protective role in axon degeneration, which runs counter to all existing data. Curiously, worm tir-1 seems to lack the NADase activity required for axon degeneration in other systems (Horsefield et al., 2019, Science). In addition, to my knowledge there is no demonstrated role for tir-1 in axon degeneration in the worm, despite axons degenerating after axotomy. Is this signaling different in this system? As a model organism biologist i hesitate to say that, but in this case it is probably true. (It is also true that ric-7 does not seem to be present in flies, zebrafish or mice. So it is hard to glean additional information from the literature on it's role in other places.)

More generally, while the authors claim loss of mitochondria is what is driving axon degeneration in ric-7, it could also be many other (perhaps totally unrelated factors) that are inducing axon degeneration. They very strongly assert this is because mitochondria are gone, which may be true, but could be wrong.

Even if correct, what if tir-1/Sarm1 need mitochondria to drive axon degeneration, but in the context of mitochondrial loss, they are unable to execute degeneration? That would not be a protective role, it would be a mitochondrially-dependent pro-degenerative requirement. This has not been tested, they are relying solely on ric-7 mutations.

I don't have much to add. My main concerns would be on potential impact. I am not convinced yet that this applies to signaling in mammals, which would be the thrust of the novelty – that in some cases SARM1 is pro-survival. Whether or not that reduces impact to the point where *eLife* would not be interested, I don't know. It is a very nice study, but I'm just not sure how to think about the result.

---

## [Author Response]

Reviewer #1:[…]Of course, because of the number of molecules involved in this pathway, there will be new things to be learned in the future. Much of my technical comments include some of the directions that I believe will be interesting. I don't see these comments as necessary to be addressed for this current manuscript except point No.1 and 2.1. To what extent, does miro-1 miro-2 double mutant still have axonal mitochondria in PVQ axon? This will help us to understand to what extent mitochondria have to be lost in order to trigger axonal degeneration.

We labeled mitochondria in *miro-1(tm1966); miro-2(tm2933)* and *miro-1(wy50180); mtx2(wy50266)* double mutants and quantified mitochondria density in axons. We found that in *miro-1(wy50180); mtx-2(wy50266)* animals, mitochondria are completely absent from axons (Figure 1—figure supplement 1B) and axons degenerate (Figure 1—figure supplement 1A). In *miro1(tm1966); miro-2(tm2933)* animals, there is a slight trend of fewer axonal mitochondria but the density is not significantly reduced compared to the control (Figure 1—figure supplement 1B), which is similar to the previous findings in touch receptor neurons (Sure et al., 2018). Therefore, the results are not sufficient to address to what extent mitochondria have to be lost in order to trigger axonal degeneration. Nonetheless, according to the previous paper ((Rawson et al., 2014), see Figure 4A-4C), it seems that one single mitochondrion can provide substantial protection against axon degeneration after injury.

We also added new data showing that the suppressors, namely the active *unc-43* mutation and *casy-1* KO, also suppress degeneration in *miro-1; mtx-2* animals, indicating that these suppressors are not specific to the *ric-7* model (We included this new data in Figure 1—figure supplement 3 and updated the text).

2. Since there are many phenotypes in kinesin 1 or klc-2 mutants, the changes of casy-1 distribution in the klc-2 is not the strongest argument for direct transport of casy-1 by kinesin motor complex. The conclusions should reflect this possibility. It would be interesting to look at casy-1 trafficking in the klc-2 mutant. I feel that "demonstrate that C. elegans calsyntenin is anterogradely trafficked by kinesin, line 31" is too strong.

We thank the reviewer for this critique. All the trafficking experiments were done with overexpression, and consistent with the endogenous data in Figure 4D, overexpressed CASY-1C in *klc-2* mutants also accumulate strongly in PVQ cell bodies (See Author response image 1). As a consequence, CASY-1C abundance in PVQ axons are much lower compared to WT controls. The weak signals in axons make it very difficult for trafficking analyses. Therefore, we did not directly compare trafficking between control and *klc-2* mutants, we have revised the section titled “Anterograde traffic of Calsyntenin/CASY-1 by kinesin promotes axon degeneration” in results and the section titled “A calsyntenin/Mint/kinesin complex controls CaMKII activity and degeneration” in discussion to make sure we don’t overstate the conclusion. Nevertheless, we would like to highlight that there is a rich body of literature suggesting that calsyntenin is transported by the kinesin motor. First, the direct interaction between mouse calsyntenin and KLC-1(the mouse homolog of the worm KLC-2) was confirmed using in vitro binding assays of purified proteins (Araki et al., 2007; Konecna et al., 2006). Second, the anterograde trafficking of calsyntenin vesicles is reduced when KLC-1 is knocked down in cultured mouse neurons (Araki et al., 2007). Moreover, the two tryptophan (W) residues in the two kinesin-binding sites that are essential for KLC binding are conserved in *C. elegans* CASY-1. Importantly, the interaction between *C. elegans* CASY-1 and KLC-2 requires the two tryptophan residues (Ohno et al., 2014). In our study, we found that the casy-1 WW/AA mutant transgene no longer promotes degeneration, presumably because it can no longer bind KLC-2 (Figure 4A). Similarly, degeneration is suppressed in the *klc-2* mutant (Figure 4B). Therefore, our data is consistent with the model that CASY-1 binds to KLC and is transported by kinesin motors.

**Author response image 1. sa2fig1:** Overexpressed CASY-1C accumulates on punctated structures in PVQ cell bodies and is at very low level in axons in *klc-2(km11)* mutants. Same imaging conditions and contrasts. Scale bar = 10 μm.

3. Is there an explanation for why the trafficking parameters of casy-1 is altered in the ric-7 mutant?

We are also very intrigued by this observation, but we are not sure what the cause is. It may be related to the lack of mitochondria and thus reduced ATP production.

4. How does LIN-10 affect ca^2+^ channel localization? LIN-10, LIN-2 and LIN-7 forms a complex to localize EGF receptors. Are LIN-2 and LIN-7 involved in degeneration protection.

We thank the reviewer for these suggestions.

For the LIN-2/7/10 complex: We tested the *lin-2 lof* allele *e1309* and it does not suppress PVQ axon degeneration in *ric-7(n2657)*. Therefore, we do not think the LIN-2/7/10 complex (or EGF receptors) are involved. We included this new data in Figure 5—figure supplement 2 and updated the texts. There is substantial published data indicating that LIN-10 has functions independent of this complex (Glodowski et al., 2005; Rongo et al., 1998).

For how loss of lin-10 affects calcium channel localization: This is a great question, but the bottom line is that we were not able to get results that we are very confident in. We labeled endogenous EGL-19 with the split-GFP approach and visualized it in PVQ. However, the endogenous EGL-19 signals were very weak and we had to use high laser intensity, high exposure and binning, which increased the background and reduced the resolution. Author response image 2 is our best examples with decent EGL-19 signals. In the PVQ cell body, there seem to be more accumulated EGL-19 puncta in *lin-10 (n1853)* mutants. However, we do not know where exactly this EGL-19 is located, e.g. cell membrane vs some kind of endosomes or golgi-derived vesicles. In the distal axon, EGL-19 signals are very weak and are often barely detectable. Similar to controls, we did observe EGL-19 puncta in *lin-10* mutants, but we were unable to make a satisfactory quantitation. Further, because the head neurons are co-labelled by the sra-6 promoter, we lose single axon resolution of PVQ after it enters the nerve ring-which is presumably the synapse and calcium channel-dense region. Thus, we do not have a definitive answer of how LIN-10 affects VGCC localization in either the cell body or the distal axon, although if pushed we would favor an effect in the cell body. Further optimization of the labeling method and an expanded investigation of VGCC components will be helpful in the future to address this question. Nonetheless, in light of the role of LIN-10 in sorting and recycling of membrane receptors (Gauthier and Rocheleau, 2021; Glodowski et al., 2005; Rongo et al., 1998; Zhang et al., 2012), we think the idea that CASY-1/LIN-10 regulates VGCC localization is an interesting possibility and discussed how it may regulate the CaMKII activity in the manuscript (See the section titled L-type calcium channels link the calsyntenin/Mint/KLC-2 trafficking complex to CaMKII activity and axon protection in Discussion)

**Author response image 2. sa2fig2:** Endogenous EGL-19 in PVQ is visualized by the NATF approach. For the distal axon, the z projections are focused either on the axon before it joins the nerve ring or the nerve ring itself. White lines indicate the distal part of the PVQ axon or the axon bundles in the nerve ring. Asterisks indicate head neurons. Scale bar = 10 μm.

In addition, although we could not resolve the question of EGL-19 localization, we do include new data showing that the p38 MAPK PMK-3 and the transcription factor CEBP-1 function downstream of the CaMKII-MAPK pathway to protect against axon degeneration (Figure 2G, Figure 5E and Figure 5F). Since CEBP-1 is a transcription factor, it likely acts in the nucleus to suppress degeneration. We updated the cartoon in Figure 6F to reflect this and revised the title, abstract and text accordingly.

5. How does mitochondria distribution defect alter the Ca dynamics in the axon?

We thank the reviewer for this comment. In a pilot experiment, we tried to measure steady-state calcium levels in PVQ axons using GCaMP signals normalized to a chemical dye. However, we did not have conclusive results due to variation and background noise from surrounding tissues. We have not examined short-term calcium dynamics, i.e. in response to stimulation. Therefore, we do not currently have good evidence for how mitochondria affect Ca dynamics in axons. But we agree this is very interesting future direction.

Reviewer #2:[…]I have very few quibbles with this work1) I think the authors should be more cautious in how they interpret the issue of unc-43 localization. All enzymes need to be localized to find their substrates. The fact that ric-7 leads to a restriction of un-43 to the cell body is interesting but the fact that moving some of it to the distal axon did not rescue the degeneration I don't think speaks very deeply to whether localization matters. The fact is we don't know where un-43 needs to be, and that forcing WT un-43 to the end of the axon does not rescue things is more or less a failed experiment. A more interesting experiment would be to see how specific localizations of WT versus active unc-43 impacts degeneration.

We thank the reviewer for the constructive critique. We agree that the relocalization experiment is not definitive enough. We decided to take it out of the manuscript and no longer make the conclusion that activity matters more than localization. We think that in order to conclude whether localization matters and where CaMKII functions, it requires us to localize WT/active CaMKII into different compartments with high accuracy, e.g. to the soma exclusively or axon exclusively. For example, restricting all WT CaMKII to the soma and test if degeneration can be induced in WT controls will tell us whether it is the loss of CaMKII in the distal axon that causes degeneration. Similarly, restricting constitutively-active CaMKII to the soma and testing if degeneration can be suppressed in *ric-7* will tell us whether active CaMKII can suppress degeneration even if it is away from the degeneration site. However, we do not know a very good tool to achieve this compartmentalization. Even if there is such a tool, it will still face the uncertainty of finding and binding CaMKII substrates. Therefore, we decided not to pursue this route.

2) Related to the point above: perhaps having too much WT unc-43 in the cell body is the real problem and the active mutant simply does a better job of leaving the cell body.The issues above serve as examples of why caution should be exercised in interpreting the localization experiments.

We agree that caution should be taken and the relocalization experiment has been taken out. Nevertheless, we do not think having too much WT UNC-43 in the soma specifically causes degeneration. First, we did not observe increased UNC-43 levels in *ric-7* animals (Figure 2D-2E). Second, the relocalization experiment, though not definitive, does suggest that simply removing UNC-43 from the soma does not suppress degeneration.

3) The EGL19 link is a potentially satisfying way to link up the unc43 hit with the CASY hit. A compelling experiment would be to use crisper/Cas9 to modify the channel conductance so that it could no longer serve as the potential source of Ca to activate unc43.

We thank the reviewer for the suggestion. The *egl-19 n582* allele that we used in the study is a reduction of function mutation in the voltage-sensing S4 segment of the channel. It has been shown to reduce calcium conductance in *C. elegans* (Jospin et al., 2002). Specifically, this mutation causes slower activation kinetics of the channel, smaller ca^2+^ entry and increased threshold for ca^2+^ transients. We have not used CRISPR/Cas9 to make more mutations in the channel pore to modify channel conductance as it would presumably require techniques such as electrophysiology to confirm that they work. We think our genetic epistasis data supports the model that ca^2+^ entry through EGL-19 activates UNC-43.

Reviewer #3:The authors sought to understand how loss of ric-7, and in turn mitochondria from axons, leads to axon degeneration. In a forward genetic screen they identify CamkII and Casy, which led to their genetic analysis of CamkII/tir-1/NSY1 and cay/mint/kin in axon degeneration. The genetic analyses are excellent, clear and well done. Loss of either of these pathway appears to suppress ric-7-induced axon degeneration. In general, the authors claims are supported by their data, although I think they need to take care in extending their claims to other systems (see below). The potential impact of the work on the field is not entirely clear to me. The major surprise would be that tir-1 (SARM1 in mammals) plays a protective role in axon degeneration, which runs counter to all existing data. Curiously, worm tir-1 seems to lack the NADase activity required for axon degeneration in other systems (Horsefield et al., 2019, Science). In addition, to my knowledge there is no demonstrated role for tir-1 in axon degeneration in the worm, despite axons degenerating after axotomy. Is this signaling different in this system? As a model organism biologist i hesitate to say that, but in this case it is probably true. (It is also true that ric-7 does not seem to be present in flies, zebrafish or mice. So it is hard to glean additional information from the literature on it's role in other places.)More generally, while the authors claim loss of mitochondria is what is driving axon degeneration in ric-7, it could also be many other (perhaps totally unrelated factors) that are inducing axon degeneration. They very strongly assert this is because mitochondria are gone, which may be true, but could be wrong.Even if correct, what if tir-1/Sarm1 need mitochondria to drive axon degeneration, but in the context of mitochondrial loss, they are unable to execute degeneration? That would not be a protective role, it would be a mitochondrially-dependent pro-degenerative requirement. This has not been tested, they are relying solely on ric-7 mutations.

We thank the reviewer for the insightful discussion and comments.

We made sure we have a concise review of the domains and functions of fly, mammalian Sarm1 and *C. elegans* TIR-1 in the Discussion section under the title “The surprising neuroprotective role of the Sarm1/TIR-1-ASK1/NSY-1 MAPK pathway” where we also discussed their similarities and differences. It is worth noting that worm TIR-1 does seem to have some NADase activity although it requires both the SAM domain and the TIR domain and has a lower activity than the human or the fly counterparts (Horsefield et al., 2019). Furthermore, induced dimerization of worm TIR-1 does lead to cell death and consumption of NAD^+^, which again is less efficient than the human counterpart (Summers et al., 2016). Despite its NADase activity, we also reviewed several lines of evidence in the discussion which suggest that the NADase activity of CeTIR-1 is not an essential factor to trigger degeneration in worms. Consistent with this hypothesis, we overexpressed worm NMNATs in *ric-7* animals and found that they do not suppress degeneration (We included this new data in Figure 2—figure supplement 2 and updated the text). This does raise the possibility that the CeTIR-1 is involved in different or additional signaling pathways. We revised this part of the discussion to better reflect the above points.

Nevertheless, we do want to highlight that all the other genes that we identified such as CaMKII and the MAPK pathway are also highly conserved and that we are examining their role in this specific degeneration model induced by the loss of mitochondria. It is possible that CeTIR-1 is versatile, and in this context exhibits a protective role through the CaMKII-MAPK branch of the signaling pathway. Our new data show that the p38 MAPK PMK-3 and the transcription factor CEBP-1 mediate the protection downstream of TIR-1, which is different from what Sarm1 does (NAD^+^ consumption) in axons. Moreover, a recent exciting paper demonstrated that active CaMKII protects against cell death in retinal ganglion cells in a variety of degeneration models in mice (Guo et al., 2021). Therefore, we agree that discrepancies exit and modified the discussion to call for more caution in extending the results, but we also think our findings will be helpful to the field.

In terms of the driving force of axon degeneration in *ric-7* mutants, we have a couple of experiments demonstrating that it is the loss of mitochondria. First, we observed no axonal mitochondria in *ric-7* as the reviewer has mentioned. Second, we examined other mitochondria trafficking mutants including *unc-116* and *miro-1; mtx-2* animals in which mitochondria are also absent and found axons degenerate similarly (Figure 2C and Figure 1—figure supplement 1A), which indicates that degeneration is not caused by some unknown defects of the *ric-7* mutation. Importantly, the suppressors, namely the active *unc-43* mutation and *casy-1* KO, also suppress degeneration in *miro-1; mtx-2* animals, suggesting that they are not specific to the *ric-7* model (We included this new data in Figure 1—figure supplement 3 and updated the text). Third, we were able to suppress degeneration by pulling mitochondria from the cell body back into the axon (Figure 1—figure supplement 1C-1D). Therefore, our observations support the model that degeneration is caused by the lack of mitochondria.

The mitochondrially-dependent pro-degenerative requirement of TIR-1 is a good alternative hypothesis for the observation that knocking out *tir-1* does not suppress axon degeneration in *ric-7* (Figure 2J). We tested this by crossing out the *ric-7* mutation and examining if overexpressing full-length tir-1 in WT animals induces axon degeneration. However, in a total of 45 animals, not a single truncated or degenerating axon was observed. Moreover, we think that the observation that overexpressing TIR-1 suppresses degeneration in *ric-7* suggests that it has a protective role at least in the absence of mitochondria (Figure 2H)-otherwise, we would have observed nothing instead of suppression.

Reference

Araki, Y., Kawano, T., Taru, H., Saito, Y., Wada, S., Miyamoto, K., Kobayashi, H., Ishikawa, H.O., Ohsugi, Y., Yamamoto, T.*, et al.* (2007). The novel cargo Alcadein induces vesicle association of kinesin-1 motor components and activates axonal transport. EMBO J *26*, 1475-1486.

Gauthier, K.D., and Rocheleau, C.E. (2021). LIN-10 can promote LET-23 EGFR signaling and trafficking independently of LIN-2 and LIN-7. Mol Biol Cell *32*, 788-799.

Glodowski, D.R., Wright, T., Martinowich, K., Chang, H.C., Beach, D., and Rongo, C. (2005). Distinct LIN10 domains are required for its neuronal function, its epithelial function, and its synaptic localization. Mol Biol Cell *16*, 1417-1426.

Guo, X., Zhou, J., Starr, C., Mohns, E.J., Li, Y., Chen, E.P., Yoon, Y., Kellner, C.P., Tanaka, K., Wang, H.*, et al.* (2021). Preservation of vision after CaMKII-mediated protection of retinal ganglion cells. Cell *184*, 4299-4314 e4212.

Horsefield, S., Burdett, H., Zhang, X., Manik, M.K., Shi, Y., Chen, J., Qi, T., Gilley, J., Lai, J.S., Rank, M.X.*, et al.* (2019). NAD(+) cleavage activity by animal and plant TIR domains in cell death pathways. Science *365*, 793-799.

Jospin, M., Jacquemond, V., Mariol, M.C., Segalat, L., and Allard, B. (2002). The L-type voltagedependent ca^2+^ channel EGL-19 controls body wall muscle function in *Caenorhabditis elegans*. J Cell Biol *159*, 337-348.

Konecna, A., Frischknecht, R., Kinter, J., Ludwig, A., Steuble, M., Meskenaite, V., Indermuhle, M., Engel, M., Cen, C., Mateos, J.M.*, et al.* (2006). Calsyntenin-1 docks vesicular cargo to kinesin-1. Mol Biol Cell *17*, 3651-3663.

Ohno, H., Kato, S., Naito, Y., Kunitomo, H., Tomioka, M., and Iino, Y. (2014). Role of synaptic phosphatidylinositol 3-kinase in a behavioral learning response in *C. elegans*. Science *345*, 313-317. Rawson, R.L., Yam, L., Weimer, R.M., Bend, E.G., Hartwieg, E., Horvitz, H.R., Clark, S.G., and Jorgensen, E.M. (2014). Axons degenerate in the absence of mitochondria in *C. elegans*. Curr Biol *24*, 760-765. Rongo, C., Whitfield, C.W., Rodal, A., Kim, S.K., and Kaplan, J.M. (1998). LIN-10 is a shared component of the polarized protein localization pathways in neurons and epithelia. Cell *94*, 751-759.

Summers, D.W., Gibson, D.A., DiAntonio, A., and Milbrandt, J. (2016). SARM1-specific motifs in the TIR domain enable NAD^+^ loss and regulate injury-induced SARM1 activation. Proc Natl Acad Sci U S A *113*, E6271-E6280.

Sure, G.R., Chatterjee, A., Mishra, N., Sabharwal, V., Devireddy, S., Awasthi, A., Mohan, S., and Koushika, S.P. (2018). UNC-16/JIP3 and UNC-76/FEZ1 limit the density of mitochondria in *C. elegans* neurons by maintaining the balance of anterograde and retrograde mitochondrial transport. Sci Rep *8*, 8938. Zhang, D., Isack, N.R., Glodowski, D.R., Liu, J., Chen, C.C., Xu, X.Z., Grant, B.D., and Rongo, C. (2012). RAB-

6.2 and the retromer regulate glutamate receptor recycling through a retrograde pathway. J Cell Biol *196*, 85-101.